# Biasing the conformation of ELMO2 reveals that myoblast fusion can be exploited to improve muscle regeneration

Viviane Tran[1,2], Sarah Nahlé[1,3,13], Amélie Robert[1,13], Inès Desanlis [1,4], Ryan Killoran[5], Sophie Ehresmann [1,3], Marie-Pier Thibault[1], David Barford [6], Kodi S. Ravichandran [7,8], Martin Sauvageau[1,2,3,9], Matthew J. Smith [5,10], Marie Kmita [1,3,4,11] & Jean-François Côté [1,2,3,4,12] ✉

Myoblast fusion is fundamental for the development of multinucleated myofibers. Evolutionarily conserved proteins required for myoblast fusion include RAC1 and its activator DOCK1. In the current study we analyzed the contribution of the DOCK1-interacting ELMO scaffold proteins to myoblast fusion. When *Elmo1*^−/− mice underwent muscle-specific *Elmo2* genetic ablation, they exhibited severe myoblast fusion defects. A mutation in the *Elmo2* gene that reduced signaling resulted in a decrease in myoblast fusion. Conversely, a mutation in *Elmo2* coding for a protein with an open conformation increased myoblast fusion during development and in muscle regeneration. Finally, we showed that the dystrophic features of the *Dysferlin*-null mice, a model of limb-girdle muscular dystrophy type 2B, were reversed when expressing ELMO2 in an open conformation. These data provide direct evidence that the myoblast fusion process could be exploited for regenerative purposes and improve the outcome of muscle diseases.

Myoblast fusion is critical for the formation of multinucleated fibers during both skeletal muscle development and regeneration[1–3]. In adults, a population of muscle-specific progenitors called satellite cells are responsible for muscle growth and regeneration[4]. In response to activating signals, satellite cells proliferate, differentiate and fuse to repair damaged myofibers[4] and the proteins and signaling pathways that control fusion are still being identified[5–21]. MYOMAKER (MYMK) and MYOMIXER (MYMX) are proteins that harbor fusogenic activity[9,13,14,18] and mutations in *MYMK* and *MYMX* result in a clinical myopathy known as Carey-Fineman-Ziter syndrome[22,23]. Similarly,

limb-girdle muscular dystrophy type 2B (LGMD2B), a myopathy resulting from mutations in *Dysferlin* (*DYSF*)[24,25], has been associated with defective membrane repair and myoblast fusion[26,27]. Other Ferlin-related proteins, such as MYOFERLIN, likewise regulate membrane resealing and cell-cell fusion[28]. In contrast to MYMK and MYMX, DYSF is not an essential component although smaller and damaged muscle fibers are observed in DYSF-null animals[27]. Since neuromuscular disorders affect millions of people worldwide[29,30], a more detailed understanding of the factors controlling myoblast fusion could contribute to the development of needed targeted therapeutics. As such,

[1]Montreal Clinical Research Institute (IRCM), Montreal, QC H2W 1R7, Canada. [2]Department of Biochemistry and Molecular Medicine, Université de Montréal, Montreal, QC H3C 3J7, Canada. [3]Molecular Biology Programs, Université de Montréal, Montréal, QC H3T 1J4, Canada. [4]Department of Medicine, Université de Montréal, Montreal, QC H3C 3J7, Canada. [5]Institute for Research in Immunology and Cancer, Université de Montréal, Montreal, QC H3T 1J4, Canada. [6]MRC Laboratory of Molecular Biology, Cambridge CB2 OQH, UK. [7]Department of Microbiology, Immunology, and Cancer Biology, University of Virginia, Charlottesville 22908 VA, USA. [8]VIB/UGent Inflammation Research Centre, Biomedical Molecular Biology, Ghent University, 9052 Ghent, Belgium. [9]Department of Biochemistry, McGill University, Montréal, QC H3G 1Y6, Canada. [10]Department of Pathology and Cell Biology, Université de Montréal, Montreal, QC H3C 3J7, Canada. [11]Department of Experimental Medicine, McGill University, Montréal, QC H3G 2M1, Canada. [12]Department of Anatomy and Cell Biology, McGill University, Montréal, QC H3A 0C7, Canada. [13]These authors contributed equally: Sarah Nahlé, Amélie Robert. ✉e-mail: jean-francois.cote@ircm.qc.ca

MYMK was used to direct heterologous reprogrammed cells to fuse with dystrophic muscles for the delivery of *Dystrophin*[31].

RAC1 and its activator DOCK1[32–42] are part of an evolutionarily conserved signaling pathway required for myoblast fusion and *Dock1* and *Rac1* were the first proteins shown to be essential for fusion in vertebrates by genetic inactivation[5,6]. Mechanistically, DOCK1 activates RAC signaling together with ELMO proteins and the activity of this complex is tightly regulated by a set of auto-inhibitory contacts[43–45]. Although the role of ELMO proteins has not yet been defined in vertebrate myogenesis, studies in *Drosophila* are consistent with ELMO playing a role in this process[46]. In the current study we demonstrate that mammalian ELMO proteins are essential for myoblast fusion and that modulating their conformation in vivo can improve muscle regeneration in response to toxin injury and the outcome of limb girdle muscular dystrophy in a murine model.

## Results

### ELMO1 and ELMO2 are essential for muscle formation

Whole-mount RNA in situ hybridization, used to define the expression profiles of *Elmo* genes in embryos at embryonic day 11.5 (E11.5), revealed *Elmo1*, *Elmo2* and *Elmo3* expression in somites which are sites of myogenesis (Supplementary Fig. 1a). *Elmo* genes were also expressed in the adult murine tibialis anterior (TA) muscle and in primary mouse myoblasts (Supplementary Fig. 1b, c). The *Elmo* genes were also found to be constitutively expressed during differentiation of cultured C2C12 myoblasts (Supplementary Fig. 1d).

Because ELMO proteins are known to interact with DOCK1/5, RAC guanine nucleotide exchange factors implicated in the control of myoblast fusion[5,47], we investigated whether ELMO1 and ELMO2 also contribute to myoblast fusion. Although *Elmo1* knockout (*Elmo1*[−/−]) mice have been reported to exhibit no gross abnormalities[48], we tested whether the expression of *Elmo2* could be compensating for this deficiency. To achieve this goal, we generated an *Elmo2*[LacZ] mouse model in which the coding sequence for ELMO2 was disrupted with the LacZ gene, whose product (ß-galactosidase) can be used as a cytomarker to assess the distribution of *Elmo2* expression (Fig. 1a; Supplementary Fig. 2a–d). *LacZ* staining of whole E11.5 embryos from mice in which one of the *Elmo2* genes was replaced with *Elmo2*[LacZ] revealed broad expression of *Elmo2*, including in somites (Supplementary Fig. 2e). Staining of E14.5 embryo sections with antibodies for ß-galactosidase and ELMO demonstrated broad expression of ELMO2 including in myosin heavy chain (MHC)-positive differentiated muscles (Supplementary Fig. 2f; Supplementary Fig. 3a).

To investigate the requirement for *Elmo1* during formation of multinucleated myofibers in vivo (Fig. 1b), we analyzed the formation of nascent myofibers in E14.5 mouse embryos and found that multinucleated myofibers were similar in wild-type (WT) and *Elmo1*[−/−] embryos (Fig. 1c). We confirmed that *Elmo1*[−/−] animals were viable and found that the cross-sectional area (CSA) of the myofibers in adult muscles was not significantly different from that in WT muscles (Supplementary Fig. 4). To explore the role of ELMO2 in myogenesis, we interbred heterozygous *Elmo2*[LacZ] mice to generate *Elmo2*-deficient animals (*Elmo2*[LacZ/LacZ]) and found that *Elmo2* was essential for embryonic development (Supplementary Table 1). Multinucleated primary muscle fibers were detected in E14.5 *Elmo2*[LacZ] embryos (Fig. 1c). To test for possible functional redundancy, we generated *Elmo1* and *Elmo2* double knockout animals, but this resulted in early embryonic lethality precluding analyses of myogenesis (Supplementary Table 1). We thus generated a conditional KO model (cKO) of *Elmo2* (*Elmo2*[flox]) (Fig. 1a; Supplementary Fig. 5a). To validate this new conditional mouse mutant, we crossed *Elmo2*[flox] mice with the global deleter *Meox2*[CRE] mice to generate an ubiquitous deletion of *Elmo2*, which resulted in embryonic lethality similar to the *Elmo2*[LacZ] mutant (Supplementary Fig. 5b; Supplementary Table 1). To achieve specific deletion of *Elmo2* in skeletal muscles, we used two independent CRE

mice, *Myf5*[CRE] and *Pax3*[CRE], which are commonly used to conditionally delete genes in the muscle lineage (Supplementary Fig. 5c). Both *Myf5*[CRE]*Elmo2*[flox/flox] and *Pax3*[CRE]*Elmo2*[flox/flox] mice were obtained at the expected Mendelian ratio and showed no obvious muscle phenotype in adults despite the absence of the *Elmo2* protein (Supplementary Fig. 3b; Supplementary Fig. 4; Supplementary Table 2). Interbreeding of *Elmo1*[−/−] mice with *Myf5*[CRE-] or *Pax3*[CRE-] *Elmo2*[flox] mice to generate double homozygous mice (*Myf5*[CRE]*Elmo1*[KO]*Elmo2*[flox/flox] and *Pax3*[CRE]*Elmo1*[−/−] *Elmo2*[flox/flox]), resulted in embryonic or perinatal lethality (Supplementary Fig. 5d; Supplementary Table 2). RT-qPCR analyses revealed that the deletion of *Elmo1* and *Elmo2* did not change *Elmo3* expression (Supplementary Fig. 5e). Molecular analyses revealed severe myoblast fusion defects in the absence of both ELMO1 and ELMO2 since mononucleated myofibers were detected (Fig. 1b, c). A reduction in muscle content was detected in E16.5 embryos, i.e. after the second wave of myogenesis, further confirming the myoblast fusion defects (Fig. 1b, d). Notably, the diaphragm was thinner in the *Elmo1/2* double homozygous mutant embryos (Fig. 1e, f) and failed to attach to the ribs (Supplementary Fig. 5f). The impaired fusion phenotype did not appear to be secondary to myogenic differentiation defects (Fig. 1b) since we detected DESMIN and MYOD expressing cells in myogenic fields in E11.5 *Elmo1/2* double mutant embryos, suggesting that differentiation was occurring normally (Fig. 1g, h; Supplementary Fig. 6a, b). We also detected the terminal differentiation marker MHC at E14.5 and E16.5 (Fig. 1c, d). Similarly, no difference in the number of mitotic or apoptotic myoblasts was observed in *Elmo1/2* double homozygous mutant embryos (Supplementary Fig. 6c–f). Collectively, these results demonstrate that ELMO1 and ELMO2 have redundant functions and are essential for embryonic myogenesis through their role in regulating primary myoblast fusion.

### Modulating developmental myoblast fusion in vivo by conformational alterations of ELMO2

We took advantage of the auto-inhibitory regulation of the ELMO/DOCK complex to create *Elmo2* mouse models that would either decrease or increase signaling output (Supplementary Fig. 7a, b; Fig. 2a)[43–45]. In the closed conformation, the N-terminal domain of ELMO (NTD), composed of a RAS-Binding Domain (RBD), an ELMO Inhibitory Domain (EID) and an ELMO domain, are in direct contact with the PH domain of ELMO and the DHR-2 domain of DOCK[44] (Fig. 2a). In the open conformation, the NTD of ELMO1 undergoes a ~120 degree rotation which relieves inhibitory contacts and exposes the RBD for binding to RHOG or ARL4A GTPases and the DHR-2 for RAC activation (Fig. 2a)[44]. To decrease signaling from ELMO2/DOCK, we generated a knock-in mouse line with a mutation in the ELMO2 RBD that abolishes its function (*Elmo2*[RBD]) (Fig. 2a; Supplementary Fig. 7a). Conversely, to increase signaling from this complex, we generated a knock-in mouse line with a mutation in the ELMO2 EID that favors its open conformation (*Elmo2*[EID]) (Fig. 2a; Supplementary Fig. 7b). The correct insertion of these mutations was extensively validated (Supplementary Fig. 7c–e). *Elmo2*[RBD/RBD] and *Elmo2*[EID/EID] mutant mice were viable and showed no gross abnormalities (Supplementary Table 3). We also confirmed that the RBD or EID mutations do not change the levels of expression of the resulting proteins (Supplementary Fig. 7f–g). As previously reported, the L43A mutation in the RBD of ELMO1 abolished its binding to RHOG or ARL4A[43,49]. To confirm that this modification had the same effect on ELMO2, WT and L43A mutant ELMO2 were purified to homogeneity and NMR spectroscopy was used to assess whether the L43A mutant had reduced affinity for RHOG. When we titrated ELMO2 WT into [15]N-labeled GMPPNP-loaded RHOG-GMPPNP, significant line-broadening was observed in the RHOG spectrum consistent with a slow tumbling rate of the large resulting complex (~105 kDa), in agreement with a bona fide interaction (Supplementary Fig. 8a). In contrast, we observed minimal line-broadening upon titration of ELMO2 L43A,

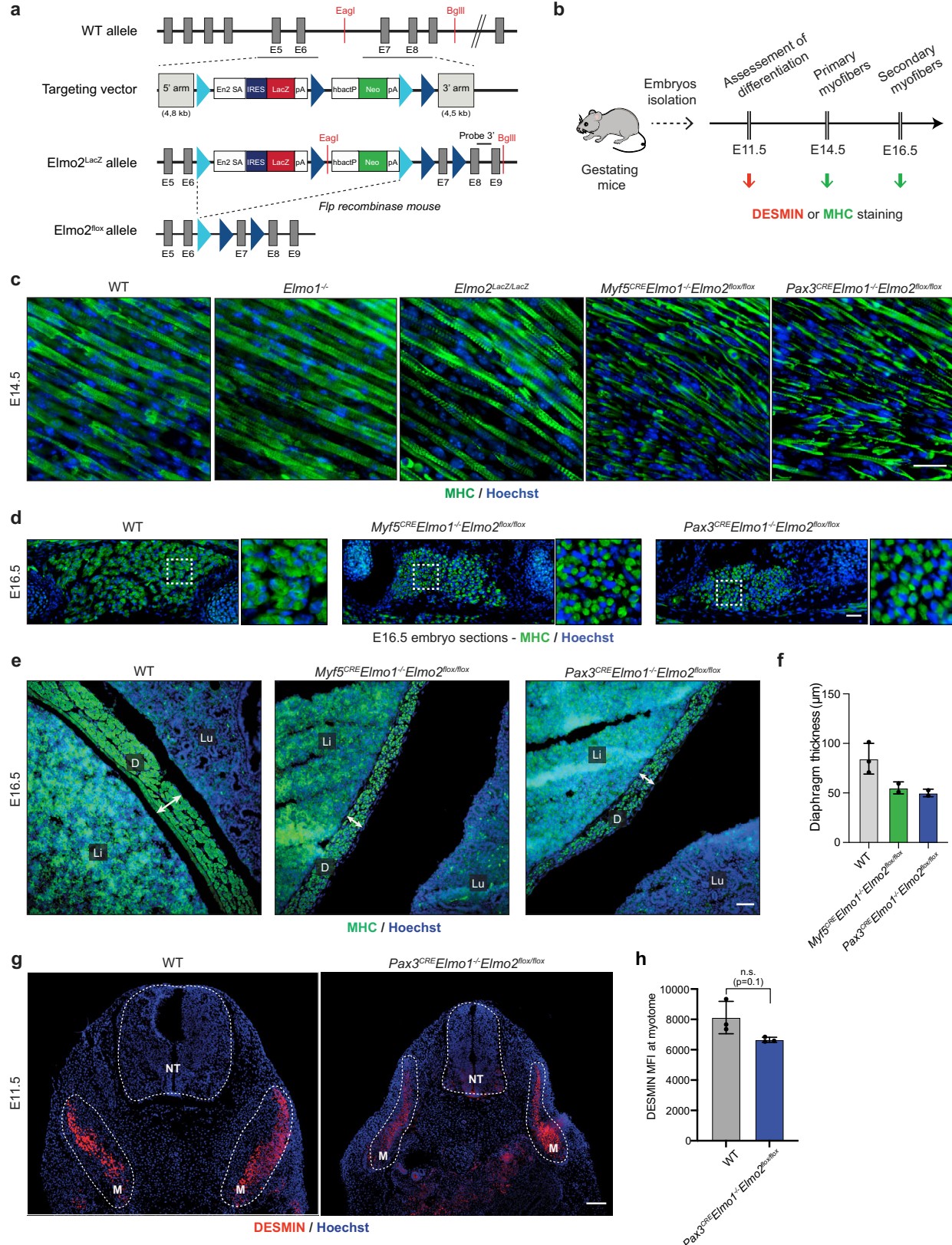

**f**

Diaphragm thickness (μm)

**h**

DESMIN MFI at myotome

suggesting that binding was disrupted (Supplementary Fig. 8b). Isothermal titration calorimetry determined a binding affinity of 10.3 μM for RHOG and WT ELMO2, while no binding could be measured for the L43A mutant (Supplementary Fig. 8c, d). These data confirm that the *Elmo2^RBD* mutant mouse expresses an ELMO2 protein with a defective RBD. We previously developed

a monomolecular biosensor of ELMO2 that demonstrated that the I196D mutation promotes an open conformation[43]. In support of this conclusion, analysing the Cryo-EM structure of the ELMO1-DOCK2 binary complex[44] indicated that substituting a charged Asp residue for the hydrophobic residue Ile204 (equivalent of I196D in Elmo2) would destabilize the closed conformation. These data

**Fig. 1 | Elmo1 and Elmo2 are essential for myoblast fusion. a** Partial representation of the *Elmo2* locus to demonstrate the strategy for the generation of *Elmo2^{LacZ}* and *Elmo2^{flox}* mice. The WT allele, the targeting vector and the recombined alleles are illustrated. The strategy and the probes used for southern blotting (Supp. Fig. 2c) are also indicated. The targeting vector contained a splice acceptor site and an IRES upstream of the *Elmo 2* exon 6, followed by the *LacZ* reporter gene. Consequently, a truncated form of *Elmo2* should be produced in fusion with IRES and *β-galactosidase*, thus resulting in a non-functional protein (*Elmo2^{LacZ}* mice). *Elmo2^{flox}* mice were generated by breeding *Elmo2^{LacZ}* with *Flp* expressing mice. En2 SA: splice acceptor of mouse Engrailed2 exon 2; IRES: internal ribosome entry site; *LacZ*: gene encoding β-galactosidase; pA: polyadenylation signal; hbactP: human β-actin gene promoter; *Neo*: neomycin resistance gene. **b** Schematic representation showing the analysis performed on mouse embryos. Differentiation of muscle cells was assessed on E11.5 embryos, primary fibers following the first wave of myogenesis were analyzed on E14.5 embryos, and secondary myofibers following the second wave of myogenesis were analyzed on E16.5 embryos. **c** Longitudinal muscle sections of E14.5 embryos. Myosin heavy chain (MHC)-positive multinucleated myofibers are present in WT, *Elmo1^{-/-}* and *Elmo2^{LacZ/LacZ}* embryos, while only mononucleated muscle cells are present in either *Myf5^{CRE}Elmo1^{-/-}Elmo2^{flox/flox}* or *Pax3^{CRE}Elmo1^{-/-}Elmo2^{flox/flox}* embryos. This experiment was done at least 3 times per genotype. **d–f** Sections of E16.5 embryos. **d** A reduction in muscle content is observed in both *Myf5^{CRE}Elmo1^{-/-}Elmo2^{flox/flox}* and *Pax3^{CRE}Elmo1^{-/-}Elmo2^{flox/flox}* embryos. This experiment was done at least two times per genotype. **e** The diaphragm thickness is smaller in both *Myf5^{CRE}Elmo1^{-/-}Elmo2^{flox/flox}* and *Pax3^{CRE}Elmo1^{-/-}Elmo2^{flox/flox}* embryos. **f** Quantification of the diaphragm thickness from **e**. Data are presented as the mean values +/− SD ($n = 3$ embryos for WT; $n = 2$ embryos for *Myf5^{CRE}Elmo1^{-/-}Elmo2^{flox/flox}* and *Pax3^{CRE}Elmo1^{-/-}Elmo2^{flox/flox}*). **g** Similar DESMIN expression is observed in control WT and *Pax3^{CRE}Elmo1^{-/-}Elmo2^{flox/flox}* embryos. **h** Quantification of DESMIN mean fluorescent intensity (MFI) at the myotome from **g**. Data are presented as the mean values +/− SD ($n = 3$ embryos). Mann-Whitney test (two-tailed) was used to determine the *p*-value. i liver, D diaphragm, Lu lung. Muscle cells and fibers were stained with anti-MHC (green) or anti-DESMIN (red), and Hoechst (blue) to reveal nuclei. (Scale bar: **c**, **d**, **e** = 50 μm; **g** = 100 μm). Source data are provided as a Source Data file.

confirm that the *Elmo2^{EID}* mouse model expresses an open conformation ELMO2 protein.

To test whether manipulating ELMO2 conformation can impact cell fusion, we analyzed the CSA of muscle fibers in 3-month-old mice and found that *Elmo2^{RBD/RBD}* mice displayed no obvious muscle phenotype, possibly due to functional compensation by ELMO1 (Fig. 2b–d). To test this, we interbred *Elmo1^{-/-}* and *Elmo2^{RBD}* mice to generate *Elmo1^{-/-}Elmo2^{RBD/RBD}* mice, which gave rise to viable offspring (Supplementary Table 3). Analysis of the myofiber CSA of these mice revealed smaller muscle fibers when compared to control mice, demonstrating that signaling of ELMO2 via its RBD is important for normal muscle development (Fig. 2b–d). To directly examine cell fusion, muscle sections from these mice were stained for DYSTROPHIN and the nuclei located inside the sarcolemma (DYSTROPHIN-stained membrane) were quantified. The number of nuclei per myofiber was decreased in *Elmo1^{-/-}Elmo2^{RBD/RBD}* mice demonstrating that less fusion has occurred during muscle development (Fig. 2f–g). In contrast, *Elmo2^{EID/EID}* mice were characterized by the presence of larger myofibers (Fig. 2b, c, e) and a higher number of nuclei per fiber was observed, demonstrating that more fusion had occurred (Fig. 2f–g). To explore whether long-term alteration in ELMO2 conformation might lead to compensating changes in gene expression, we isolated and differentiated primary myoblasts from WT and *Elmo2^{EID/EID}* mice and analysed their transcriptomes by RNA-Seq (-900 differentially expressed genes identified; Supplemental Fig. 9 and Supplemental Dataset 1), with particular attention paid to differential expression of genes coding for core myoblast fusion machinery. Among the 30 genes associated in the literature with myoblast fusion, 27 were expressed in our myoblasts and the expression of three (*PAK1*, *TGFBR2*, *DYSF*) was altered (Fig. 2h; Supplementary Fig. 9d). Notably, the expression of the fusogenes *MYMK* and *MYMK* were not affected. Collectively, these results demonstrate that controlling the signaling output of the ELMO/DOCK complex through manipulation of the conformation of ELMO2 impacts skeletal muscle development.

### Modulating muscle regeneration in vivo by conformational changes of ELMO2

These results led us to investigate whether the myoblast fusion process could be manipulated for regenerative purposes. To this end, regeneration of tibialis anterior (TA) muscles was analyzed by measurement of muscle cross-sectional area (CSA) in *Elmo2^{RBD/RBD}* and *Elmo2^{EID/EID}* mice at 7, 14 and 21 days following cardiotoxin (CTX)-induced injury. While *Elmo2^{RBD/RBD}* mice presented a normal muscle regeneration profile in comparison to WT or *Elmo1^{-/-}* mice, the CSA of regenerated muscle fibers from *Elmo1^{-/-}Elmo2^{RBD/RBD}* mice revealed a decrease in myofiber size at both 14 and 21 days following CTX injections, demonstrating that muscle regeneration was less efficient when

signaling by the RBD of ELMO2 was abolished (Fig. 3a–c). Conversely, *Elmo2^{EID/EID}* mice exhibited larger regenerated muscle fibers at both 14 and 21 days following the CTX-induced injury (Fig. 3a, d–e). Strikingly, a higher number of myofibers with three nuclei or more was observed in *Elmo2^{EID/EID}* mice, demonstrating that more myoblast fusion events occurred in the regenerating muscles of these mice (Fig. 3f). Notably, 7 days following CTX-induced injury the measured fibers size and number of nuclei per fiber was similar between WT and *Elmo2^{EID/EID}*, suggesting that increased myoblast fusion in mice expressing ELMO2^{EID} occurs after this stage (Supplementary Fig. 10a–d). Besides, post-injury inflammatory response seemed unaffected by ELMO2^{EID} as depicted by the equivalent infiltration of F4/80+ macrophages at the injection site 3 days post injury in *Elmo2^{EID/EID}* and control mice (Supplementary Fig. 10e). To investigate whether expression of ELMO2^{EID} is sufficient to enhance fusion in the absence of ELMO1, we interbred *Elmo1^{-/-}* with *Elmo2^{EID/EID}* mice. However, viable *Elmo1^{-/-} Elmo2^{EID/EID}* offspring could not be recovered (Supplementary Table 3). These data demonstrate that the autoinhibition activity of the ELMO/DOCK proteins is essential for embryonic development. Although E14.5 and E16.5 embryos were grossly abnormal (Supplementary Fig. 11a), skeletal muscle from E14.5 embryos showed no obvious defects and fusion occurred normally when only ELMO2^{EID} was expressed (Supplementary Fig. 11b). Finally, the number of PAX7-positive satellite cells was similar in *Elmo2^{EID/EID}* and *Elmo1^{-/-}Elmo2^{RBD/RBD}* mice, making it unlikely that a change in the available stem cell pool was responsible for the observed variations in fiber size seen in these models (Fig. 3g–h). These results demonstrate that manipulation of the conformation of ELMO2 impacts skeletal muscle regeneration.

### ELMO2 conformation mutants impact fusion in a myoblast-intrinsic manner

Because the knock-in mouse models of *Elmo2* produce ELMO2^{RBD} or ELMO2^{EID} proteins in the whole organism, we sought to determine whether the observed effects on cell fusion in vivo were the result of myoblast-intrinsic mechanisms. To test this, we isolated primary myoblasts and assessed differentiation and fusion in vitro. Our results revealed decreased fusion in primary myoblasts derived from *Elmo1^{-/-}Elmo2^{RBD/RBD}* mice as indicated by an increase in mono-nucleated MHC-positive fibers with a concomitant decrease in fibers of three or more nuclei (Fig. 4a, b). In contrast, myoblasts isolated from *Elmo2^{EID/EID}* mice showed evidence of increased fusion activity characterized by an increase in MHC-positive myofibers containing three nuclei or more and a decrease in mononucleated cells (Fig. 4a, b). Since *Elmo2^{EID/EID}* mice have larger fibers during muscle development and regeneration, we investigated whether the *Elmo2^{EID}* myoblasts have an increased capacity for myoblast-myotube fusion. We performed a mixed cell population assay in which isolated WT cells stained with a live cell

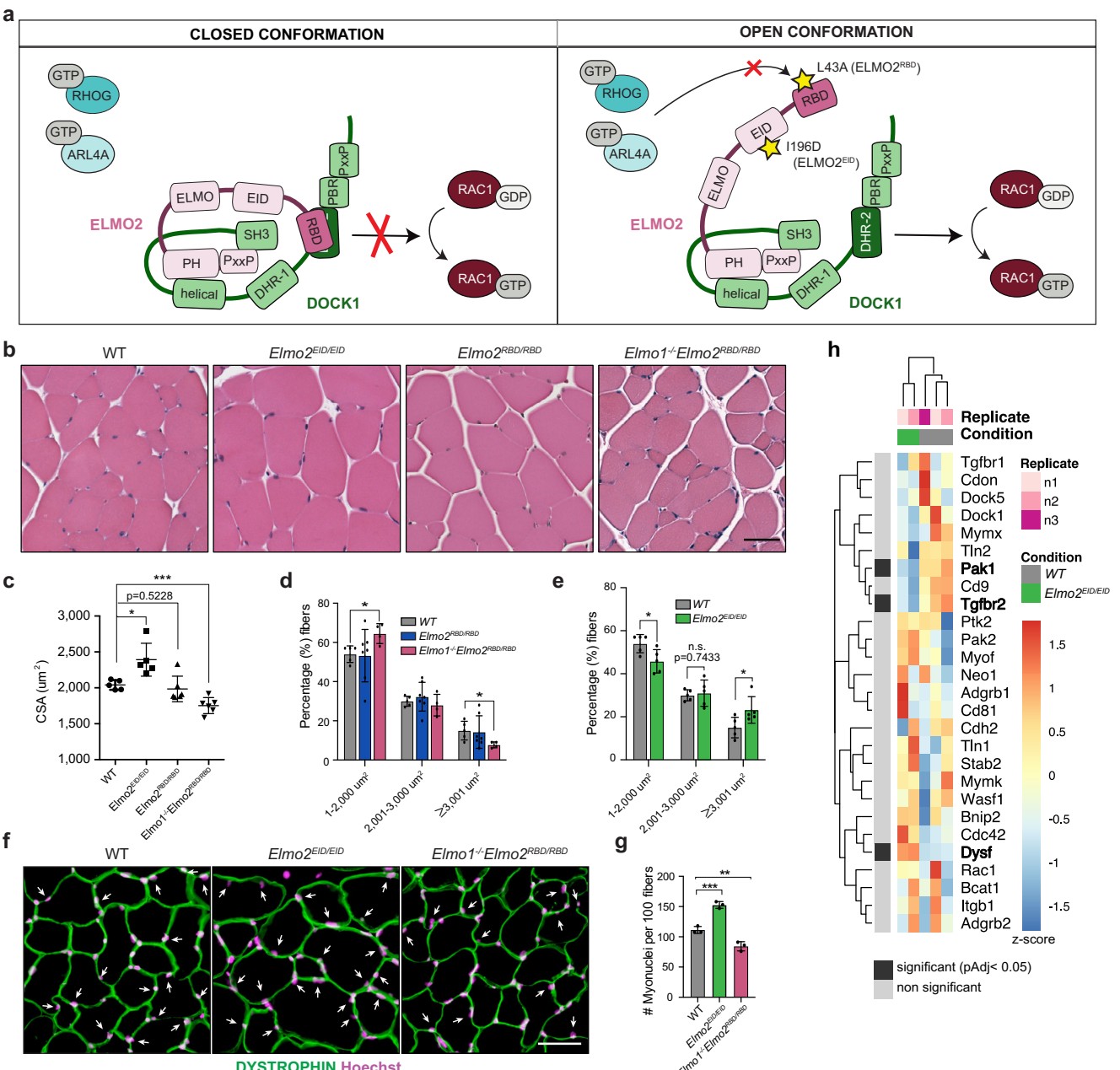

**Fig. 2 | Manipulating ELMO2 conformational regulation impacts on myoblast fusion during muscle development and growth. a** Schematic representation showing the closed and open conformations of the ELMO·DOCK complex. In the closed state, ELMO is in a closed conformation and the DHR-2 domain of DOCK is blocked by the RBD domain of ELMO, thus preventing RAC1 activation and the binding of interactors to the RBD of ELMO. Upon activation of the ELMO-DOCK complex, binding sites for ELMO interactors become available and the DHR2 of DOCK can activate RAC1. The RBD L43A mutation abrogates the binding of the RHOG and ARL4a GTPases, hence diminishing the signaling from ELMO/DOCK. The EID I196D mutation favors the open conformation of ELMO, which increases RAC1 activation by ELMO/DOCK. **b** Representative muscle fibers cross-sections of the indicated mice stained with H&E. **c** Quantification of **b** showing the mean myofibers cross-sectional area (CSA) +/−SD per genotype (*n* = 5 mice). **d**, **e** Distribution of myofibers size from (**b**) of the indicated mice. Data are presented as the mean

values +/− SD (*n* = 5 mice). **f** Cross-sectional muscle sections of WT, *Elmo2^EID/EID* and *Elmo1^−/−^Elmo2^RBD/RBD* mice stained with anti-DYSTROPHIN (green) and Hoechst (magenta). Arrowheads indicate myonuclei located inside the myofibers. **g** Quantification of the number of myonuclei located inside the myofibers of **e**. Data are presented as the mean values +/− SD (*n* = 3 mice). (Scale bar: 50 μm). The Student's *t*-test (for comparison of two independent groups) was used to calculate the *P*-values (two-tailed) presented in **c**–**e**, **g**; *\*P* < 0.05, *\*\*P* < 0.01, *\*\*\*P* < 0.001.
**h** Expression heatmap of known fusogenes between WT and *Elmo2^EID/EID* myoblasts. Samples and genes were clustered using Euclidian distance. *P*-values for differential expression were calculated using the Wald test and adjusted for multiple comparisons using the Benjamini-Hochberg method. Significantly differentially expressed genes (padj < 0.05) are represented by black boxes. Source data are provided as a Source Data file.

tracking dye (red−pseudo colored magenta) were differentiated into myotubes and subsequently mixed with myoblasts from either WT or *Elmo2^EID/EID* mice that were stained with a green tracking dye (Fig. 4c). In this approach, the appearance of light pink myofibers are indicative of

a fusion between the magenta myotubes and the green myoblasts (Fig. 4c). When *Elmo2^EID* myoblasts were mixed with WT myotubes, an increased proportion of light pink fibers was observed as compared to the same experiment using control myoblasts, demonstrating that

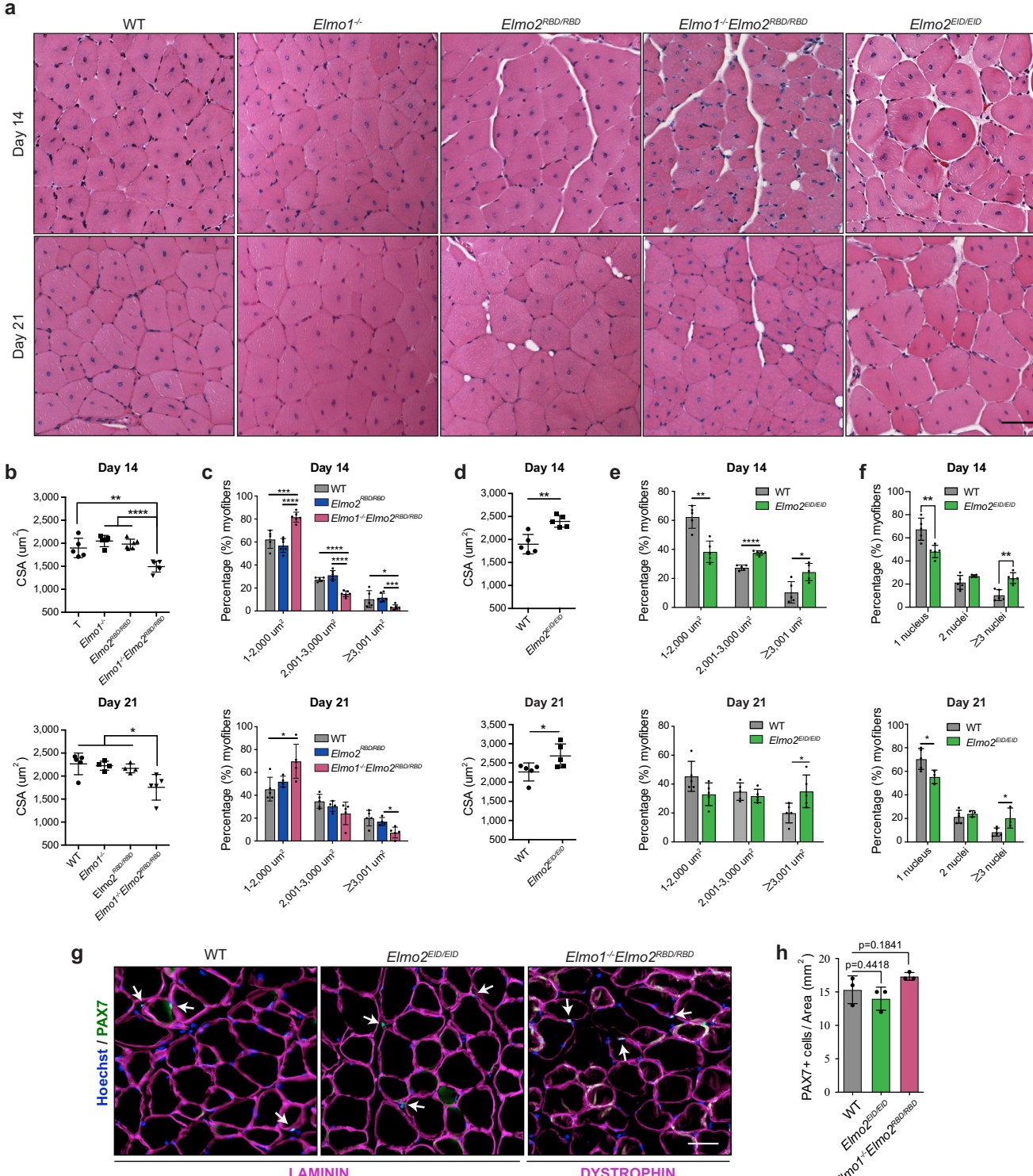

**Fig. 3 | Manipulating ELMO2 conformational regulation impacts on myoblast fusion during muscle regeneration. a** Representative cross-sections of TA muscle 14 and 21 days following CTX-induced injury. **b–f** Quantification of (**a**) showing **b**, **d** the mean CSA per mice, **c**, **e** the percentage (%) of myofibers at different ranges of myofiber size and **f** the number of nuclei per myofibers at day 14 and 21 following injury for the indicated mice. Data are presented as the mean values +/− SD ($n = 5$ mice). **g**, **h** The number of PAX7-positive cells is not affected in both *Elmo2^{EID/EID}* and *Elmo1^{-/-}Elmo2^{RBD/RBD}* mice. Satellite cells are stained with anti-PAX7 (green), the basement membrane of muscle is stained with anti-LAMININ or anti-DYSTROPHIN (magenta) antibodies and nuclei were revealed with Hoechst (blue). (Scale bar: 50 μm). **h** Quantification of (**g**) Data are presented as the mean values +/− SD ($n = 3$ mice) The Student's *t*-test (for comparison of two independent groups) was used to calculate the *P*-values (two-tailed) in **b–f**, **h**; *$P < 0.05$, **$P < 0.01$, ***$P < 0,001$ and ****$P < 0.0001$. Source data are provided as a Source Data file.

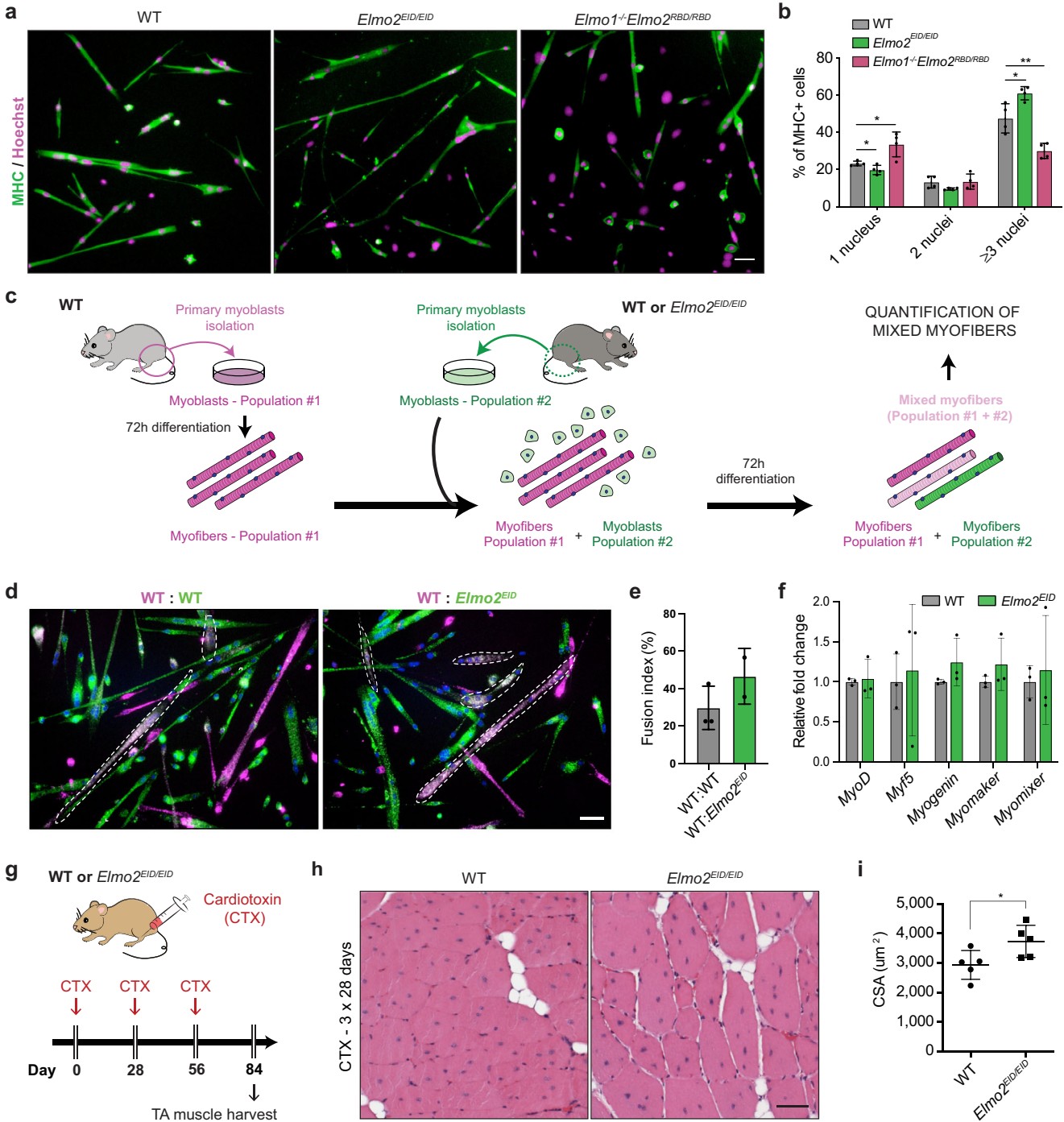

**Fig. 4 | Manipulating ELMO2 conformational regulation regulates myoblast fusion in a cell intrinsic manner. a** Fusion assay performed with primary myoblasts isolated from the indicated mice after 72 h of differentiation. Myoblasts were stained with anti-MHC (green) and Hoechst (magenta) to reveal the nuclei ($n = 4$). **b** Quantification of (**a**) showing the fusion index of mononucleated (1 nucleus), binucleated (2 nuclei) and multinucleated myofibers (≥3 nuclei). Data are presented as the mean values +/− SD ($n = 4$). **c** Schematic representation of the mix population assay performed with primary myoblasts isolated from the indicated mice. The first population of primary myoblasts were stained with a red dye (represented here in magenta) and were differentiated for 72 h. A second population of primary myoblasts of the indicated genotypes were stained with a green dye and added with the magenta myofibers. A second round of differentiation was induced for 72 h and light pink myofibers (mixed myofibers) were quantified. **d** Mix population assay performed with primary myoblasts, as described in **c**. **e** Quantification of (**d**)

showing the fusion index of light pink myofibers (mixed myofibers) for each condition. Data are presented as the mean values +/− SD ($n = 3$ independent experiments for WT:WT and $n = 2$ independent experiments for WT:Elmo2$^{EID}$). **f** Relative fold change from the indicated differentiation markers were obtained from RNA sequencing performed on WT or *Elmo2$^{EID}$* myoblasts differentiated for 72 h. Data are presented as the mean values +/− SD ($n = 3$). **g** To induce multiple cycles of degeneration/regeneration, CTX was injected in the TA 3 times, every 28 days. One month following the last injection, muscle was harvested and analyzed. **h** Representative muscle cross-sections stained with H&E of mice of the indicated genotype following repeated CTX injections. **i** Quantification of (**g**) showing the mean CSA per mice. Data are presented as the mean values +/− SD ($n = 5$ mice). (Scale bar: 50 μm). Student's *t*-test (for comparison of two independent groups) was used to calculate the *P*-values (two-tailed) presented in **b** and **i**; *$P < 0.05$, **$P < 0.01$. Source data are provided as a Source Data file.

more myoblasts fused when the active form of ELMO2 was expressed (Fig. 4d, e). No change in the mRNA levels of differentiation markers was detected in the myoblasts expressing the open conformation form of ELMO2 (Fig. 4f). *MYMK* and *MYMX* mRNA levels, as assessed by RT-qPCR, were also unchanged following differentiation of *Elmo2*[EID] myoblasts (Fig. 4f), suggesting that the increase in myoblast fusion observed in these cells was not a result of increased expression of the fusogens. Moreover, we detected no difference in actin organization at different stages of differentiation of WT and ELMO2[EID] myoblasts (Supplementary Fig. 12a–f). Given the importance of RAC1 signaling in cell migration, we investigated by live imaging whether cell motility alterations could explain the differences in cell fusion and no differences were found in either speed or directionality between WT primary myoblasts or cells isolated from *Elmo1*[−/−]*Elmo2*[RBD/RBD] or *Elmo*[EID/EID] mice (Supplementary Fig. 12g–h). Collectively, these results demonstrate that manipulating the conformational state of ELMO2 directly modulates the fusion capability of myoblasts, but not their differentiation or migration, in vitro.

Increase in fusion for regenerative or therapeutic purposes should not come at the expense of depleting satellite cells. To mimic chronic muscle diseases where myofibers undergo cycles of degeneration and regeneration and to test for the effect of ELMO2-activated myoblast fusion on the available pool of satellite cells, we conducted multiple CTX injection/regeneration assays in *Elmo2*[EID/EID] mice (Fig. 4g). Analysis of the tibialis anterior (TA) muscles from *Elmo2*[EID/EID] mice revealed the presence of larger fibers even after several rounds of muscle injury (Fig. 4h, i). These results demonstrate that the pools of satellite cells required for muscle injury repair are maintained in *Elmo2*[EID/EID] mice, allowing for sustained improvements in muscle regeneration.

### ELMO2-mediated increase of myoblast fusion improves the dystrophic features of DYSFERLIN-null mice

Carey-Fineman-Ziter syndrome is caused by rare inherited myopathies implicating genes involved in myoblast fusion[22]. Mutations that weaken the function of the myoblast-specific fusogens MYMK and MYMX are the most direct examples of such genes. Although there are no existing animal models that mimic Carey-Fineman-Ziter syndrome resulting from MYMK mutations, mutations in *DYSF*, the gene responsible for heterogenous limb-girdle muscular dystrophy type 2B (LGMD2B), results in myoblast fusion deficiency. LGMD2B patients are predominantly affected in the proximal muscles of the limbs and trunk[50]. Primary myoblasts derived from LGMD2B patients have low MYOG expression level, but also show severe myoblast fusion defects as mostly binucleated fibers were detected in vitro[26]. Accumulation of DYSF at the growing end of the myofibers, where myoblasts are fusing, suggests a function for the protein during fusion[26]. Indeed, *Dysf*-null mice exhibit many of the key features seen in LGMD2B patients, including progressive muscular dystrophy, myofibers with centrally localized nuclei, muscle necrosis, fat replacement and infiltration of macrophages[27,51]. Like myoblasts derived from LGMD2B patients, *Dysf*-null myoblasts display defective fusion in vivo and in vitro[27]. Myofibers from *Dysf*-null mice present a reduced CSA, and although explanted myoblasts can form binucleated myofibers, they are unable to form larger fibers through fusion of additional myoblasts, suggesting that DYSF regulates myoblast to myotube fusion[27]. Hence, *Dysf*-null mice represent a valid pre-clinical model in which the value of increasing myoblast fusion efficiency in disease outcome can be tested.

To this end, *Dysf*[−/−] and *Elmo2*[EID/EID] mice were crossed and the resulting *Dysf*[−/−]*Elmo2*[EID/EID] animals were challenged with CTX-injections to induce tissue regeneration and to test myoblast fusion. Analysis of myofibers CSA at 14 days following CTX-induced injury demonstrated that *Dysf*[−/−] mice display smaller myofibers (Fig. 5a, b). Strikingly, the myofiber CSA was restored and similar to that seen in control animals in *Dysf*[−/−]*Elmo2*[EID/EID] mice, suggesting that ELMO2 in an open conformation can rescue the myoblast fusion defects due to the absence of DYSF (Fig. 5a, b). PAX7 staining and quantification demonstrated that the observed phenotypes were not linked to a variation in the number of stem cells (Fig. 5c, d). We also verified that ELMO2 expression was similar in WT, *Dysf*[−/−] and *Dysf*[−/−]Elmo2[EID] muscles (Fig. 5e). To determine whether the ELMO2[EID]-dependent increase in myoblast fusion corrected additional features of the disease as it progresses, we analyzed the quadricep muscles of aging mice (8–12 months old). As expected, the muscle from *Dysf*[−/−] mice displayed myofibers with centrally located nuclei[27,52], in agreement with the notion that these muscle fibers are constantly undergoing repeated cycles of degeneration/regeneration (Fig. 5f, g). Moreover, necrotic myofibers and fatty deposits were observed in these mice (Fig. 5f, h–i). Interestingly, expression of the active form of ELMO2 in the *Dysf*[−/−] mice led to a decrease in the number of myofibers with centrally located nuclei as well as the amount of necrotic myofibers (Fig. 5g–h). In contrast, the abnormal fatty deposits were not rescued in the *Dysf*[−/−]*Elmo2*[EID/EID] (Fig. 5f, i). To determine whether the open conformation ELMO2 could directly rescue the fusion defects of *Dysf*-null myoblasts, we performed in vitro experiments and found that cells isolated from *Dysf*[−/−]*Elmo2*[EID/EID] mice displayed an increased capacity to fuse as compared to *Dysf*[−/−] mice (Fig. 5j, k). These results directly demonstrate that increasing the activity of the ELMO2/DOCK complex through a conformational change in ELMO2 can improve the efficiency of myoblast fusion in both normal animals as well as in models of clinically relevant diseases where fusion is impaired.

## Discussion

The fusion of myoblasts relies on at least two critical mechanisms: the expression of fusogenic proteins and actin dynamics[9,12–14,53–56]. How these two aspects are interconnected remains to be determined[57] and the exact contribution of ELMO/DOCK-mediated actin dynamics to fusion has not yet been deciphered. Our work provides an important conceptual advance in understanding the process of myoblast fusion by demonstrating an essential contribution of both ELMO1 and ELMO2 proteins to embryonic myoblast fusion, revealing their evolutionarily conserved role in this process[46]. We found that mice lacking both ELMO1 (globally) and ELMO2 (muscle specific) died at birth, and we speculate that this is secondary to respiratory failure as a result of poorly developed muscles. At this time, we don't know the cause of death following global ELMO2 elimination and further study of this mouse model may reveal additional biological functions unique to ELMO2.

We also established that we can genetically modulate the signaling output of the ELMO2/DOCK complex by altering the conformational state of ELMO2 in mice. With this approach, we were able to decrease ELMO2-mediated signaling by inactivating its RBD, therefore uncoupling the ELMO2/DOCK complex from RHOG and ARL4A, leading to decreased fusion in vivo during development and regeneration, as well as in vitro. Conversely, in similar assays, we could increase myoblast fusion both in vivo and in vitro by stabilizing ELMO2 in an open conformation. Mechanistically, the increase in fusion mediated by ELMO2[EID] is independent from the changes in the expression levels of the fusogenes (MYMK and MYMX). Comparing the expression levels of genes known to be part of the core machinery of myoblast fusion revealed changes in only *Pak1*, *Tgfbr2* and *Dysf* (Fig. 2h). Recent studies demonstrated that TGFBR2 contributes to pace the speed of myoblast fusion[20,21] and accordingly, the decreased expression of this gene in ELMO2[EID] myoblasts is consistent with increased fusion. The increased expression of *Dysf* in *Elmo2*[EID] myoblasts is interesting and warrants further mechanistic studies. However, we found that ELMO2[EID] rescues fusion of *Dysf*-null myoblasts suggesting that upregulation of DYSF is not essential for the pro-fusion effects of this protein. In WT myoblasts, it is possible that ELMO proteins act in concert with DYSF. Collectively, these results suggest a direct role of the ELMO2[EID] mutation on RAC-controlled signaling events, although we cannot exclude functionally

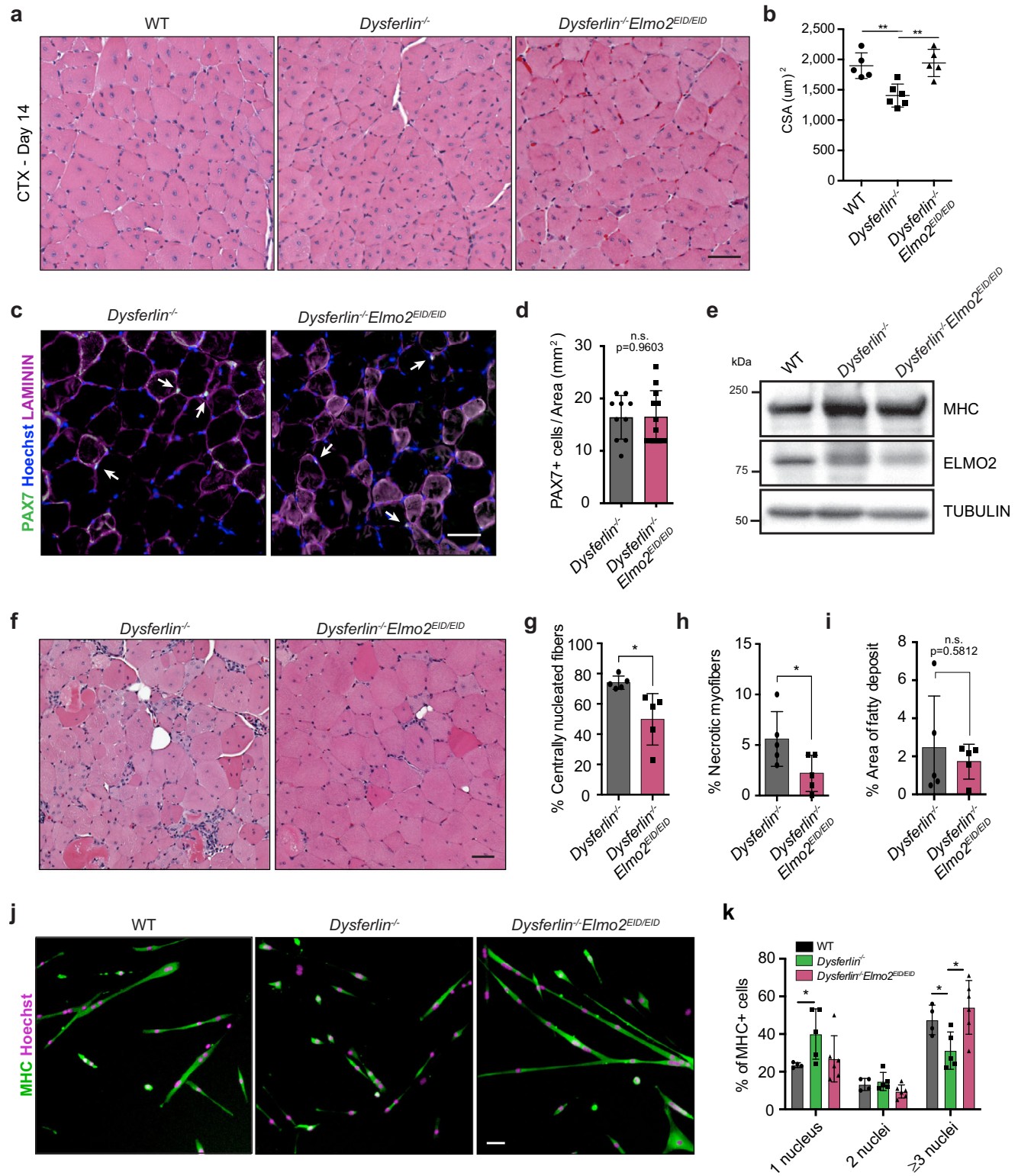

important transcriptional changes. We found that mice lacking ELMO1 and expressing ELMO2[RBD] demonstrate muscle regeneration defects, but are otherwise apparently normal. This reveals that the RBD domain in these proteins is not vital to their functions. Conversely, mice lacking ELMO1 but expressing ELMO2[EID] displayed severe lethal developmental abnormalities. It is possible that this modification could either cause excessive RAC1 activation that disturbs embryogenesis or that this "activating" mutation also leads to the loss of some of ELMOs' other functions. We strategically engineered mice

with RBD or EID mutations in the endogenous gene locus to preserve normal temporal expression and physiological levels of the resulting ELMO2 mutant proteins. A limitation of this approach is that additional cell types may be affected by ELMO2 conformational changes, such as alterations in the recruitment of immune cells to injured muscles. However, experimentally, we found that macrophages were recruited similarly to the muscles of control or *Elmo2[EID]* mice following CTX injection. Our in vitro experiments further confirmed our hypothesis that changes in ELMO2 conformation modulate myoblast fusion.

**Fig. 5 | Expression of open conformation of ELMO2 rescues the dystrophic phenotypes of the *Dysferlin*−/− mouse model. a** Representative muscle cross-sections of the indicated mice stained with H&E, at day 14 following CTX-injection. **b** Quantification of (**a**) showing the mean CSA per mice +/−SD (*n* = 5 mice). WT mice used for quantification are the same group as in Fig. 2g–l. **c** Representative muscle cross-sections of mice of the indicated genotypes stained with anti-LAMININ(magenta), anti-PAX7 (green) and Hoechst (blue). Arrowheads indicate PAX7-positive cells located on the myofibers. **d** Quantification of the number of PAX7-positive cells from (**c**). Data are presented as the mean values +/− SD (*n* = 10 measurements for Dysferlin−/−, *n* = 12 for Dysferlin−/−Elmo2EID/EID, from two mice per condition). **e** Western blot analysis of ELMO2 expression in primary myoblast isolated from the indicated mice genotypes. Data are representative from two independent mice per condition. **f** Representative cross-sections of myofibers stained with H&E, from aged (8–12 months old) mice of the indicated genotypes (*n* = 5 mice). **g**–**i** Quantification of the muscle cross-sections from (**f**) of the percentage of the centrally located nuclei myofibers (**g**), necrotic myofibers (**h**) and area of fatty deposit (**i**). Data are presented as the mean values +/− SD (*n* = 5). **j** Fusion assays performed on primary myoblasts isolated from mice of the indicated genotypes (*n* = 5). **k** Quantification of (**j**) showing the fusion index of MHC-positive myofibers. Data are presented as the mean values +/− SD (*n* = 5). WT myoblasts used for quantification are the same cells as in Fig. 4a, b. (Scale bar: 50 μm). Student's *t*-test (for comparison of two independent groups) was used to calculate the *P*-values (two-tailed) in **b**, **d**, **g**–**i**, **k**; *P* < 0.05, **P* < 0.01. Source data are provided as a Source Data file.

These results strongly support a myoblast-specific function of ELMO2EID in the increased regeneration phenotypes observed in this study.

Taken together, our data demonstrate that the ELMO/DOCK pathway could be exploited for regenerative therapies by using the conformation of ELMO2 to modulate RAC1 signaling. Superimposing the ELMO2-BAI1 complex[58] onto the ELMO1 subunit of the DOCK2-ELMO1[44] complexes indicated that the BAI receptor-binding site on ELMO1 is accessible only in the active ternary DOCK2-ELMO1-RAC1 complex (Supplementary Fig. 13a). In the inactive binary DOCK2-ELMO1 complex, ELMO1 adopts a closed autoinhibited conformation that buries the BAI receptor-binding site. In the binary state, residues of the BAI receptor-binding site form intramolecular interactions with the PH domain of ELMO1 (ELMO1PH) (Supplementary Fig. 13b). Thus, BAI peptides binding to this site would release the ELMO1-induced auto-inhibition, thereby stabilizing the active conformation of DOCK2-ELMO1, and could be exploited for regenerative strategies.

ELMO/DOCK signals downstream of the G-protein-coupled receptors BAI1 and BAI3 in myoblast fusion[8,10,15]. It is possible that the open conformation of ELMO facilitates binding to these receptors, but we have not been able to demonstrate this. Alternatively, since individual knockouts of BAI1 and BAI3 have relatively modest myogenesis phenotypes, we believe that there may be additional cell surface proteins that engage ELMO/DOCK to promote cell fusion. As such, it will be important to define the repertoire of ELMO-DOCK-associated proteins in myoblasts undergoing fusion to fully explain how this complex functions.

ELMO proteins have also been reported to contribute to Cadherin-mediated adherent junctions in MDCK epithelial cells and in *Drosophila* S2 cell[59]. Hence, one hypothesis is that ELMO-null cells have trouble establishing the strong cell-cell junctions needed for fusion. This should be explored in future studies. Alternatively, ELMO2EID myoblast could have increased cell-cell contacts. We explored the differentially expressed genes between WT and ELMO2EID myoblasts and found an enrichment in the GO term "cell-cell adhesion" (Supplementary Fig. 9c). From this, six genes were up regulated in *Elmo2EID* myoblasts (*Dchs1*, *Lims2*, *Robo4*, *Esam*, *Perp* and *Itga2*), but none of them have yet been directly implicated in cell fusion.

We established that the open conformation of ELMO2 protein provides a therapeutic opportunity in a disease context. We revealed that ELMO2EID corrects several features of the LGMD2B model (*Dysf*−/− mice). Notably, ELMO2EID rescued the fusion defects of *Dysf*-null myoblast in vivo and in vitro. This is a major advance since no therapies are currently available for this debilitating disease. Whether this will be applicable to other fusion diseases or muscular dystrophies that show extensive degeneration/regeneration features remains to be investigated. Controlling the conformational state of ELMO2 establishes the concept that increasing myoblast fusion efficiency provides a therapeutic opportunity in muscle diseases. Likewise, it can be broadened to other situations where muscle regeneration is affected, such as during physical activity, cachexia or aging. Hence, further

understanding of the molecular events controlling this critical cell fusion event may reveal additional therapeutic targets to manipulate and improve myoblast function.

## Methods

### Muscle regeneration−cardiotoxin injury
Experiment was performed as previously described[15]. Briefly, mice were anesthetized by isoflurane inhalation. 50ul of cardiotoxin (CTX, stock concentration: 10 μM) from *Naja mossambica mossambica* (Sigma) was injected in the TA muscle of mice, with a 26-gauge needle. The TA muscles were collected at the indicated days following CTX-induced injury and processed for analysis by histology, as described below.

### Histology and myofibers CSA analysis
Adult muscles were dissected, fixed with 10% formalin and embedded into paraffin blocks (according to standard procedures). 5μm sections were obtained and stained with hematoxylin and eosin (H&E). Pictures of muscle sections were captured using Zeiss Axiophot microscope (between 5 and 10 pictures per mouse genotype). Quantification of the CSA of myofibers was done using the Volocity software (PerkinElmer Life and Analytical Sciences). More precisely, the outline of each single myofiber in one picture was drawn manually, for the Volocity software to calculate the area of the muscle fiber. For each mouse, the mean of the myofiber CSA was calculated and the value was used for the generation of the graphs using the Prism Graph software.

### Antibodies
Monoclonal antibodies anti-PAX7 (1:10) and anti-MYOSIN HEAVY CHAIN (MHC - MF20) (1:20) were obtained from the Development Studies Hybridoma Bank (Iowa City, IA) (Cat. Pax7 and MF20). Mouse monoclonal anti-DESMIN (1:100) was obtained from Sigma (Cat. D1033). Mouse monoclonal anti-MYOD (clone 5.8 A, 1:50) was obtained from BD Biosciences (Cat. 554130). Anti-DYSTROPHIN antibody (1:200) was obtained from Abcam (ab15277). Anti-phospho HISTONE-H3 (Ser-10) antibody (1:100) was obtained from Cell Signaling (#9701). Anti-LAMININ DyLight 650 antibody (1:250) was obtained from Novus (Cat. NB300-144C). Anti-ß-Galactosidase (1:100) was obtained from Thermo Fisher (A-11132). Alexa Fluor 488-conjugated-Rat monoclonal anti-F4/80 (clone A3-1, 1:100) was from Biorad (Cat. MCA497A488T). Rabbit polyclonal antibody against Elmo was custom generated (Genscript), and the antibody recognized both ELMO1 and ELMO2. Alexa Fluor 488 chicken anti-mouse IgG, Alexa Fluor 488 chicken anti-rabbit IgG, Alexa Fluor 568 goat anti-mouse IgG and Alexa Fluor 568 anti-rabbit (1:1000) were obtained from Thermo Fisher.

### Immunohistochemistry−embryo sections
For MHC, ß-Galactosidase and ELMO staining, experiment was performed as previously described[5]. Briefly, embryos were fixed in 4% paraformaldehyde for 1 h and incubated in 20% sucrose/PBS overnight. 10μm cryosections of embryos were obtained and antigen retrieval

protocol (10 mM sodium citrate buffer, pH 6) was performed. Following blocking in 5% BSA/0.1% Tween-20/PBS for 1 h, embryo sections were incubated with primary antibody overnight. The next day, sections were incubated with secondary anti-mouse (MHC) or anti-rabbit (ß-Galactosidase and ELMO) antibody for 1 h and with Hoechst for 10 min. TUNEL kit was used to detect apoptotic cells, according to the manufacturer's protocol. Anti-phospho-HISTONE-H3 antibody was used to detect mitotic cells. For MYOD and DESMIN staining, embryos were fixed in 4% paraformaldehyde for 5 min and incubated in 20% sucrose/PBS overnight. 10 µm sections were obtained from OCT: 20% sucrose/PBS (1:1) embedded embryos and permeabilized with 0.2% Triton/PBS. Following blocking in 1%BSA/PBS for 1 h, embryo sections were incubated with primary antibody (anti-MYOD or anti-DESMIN) overnight. The next day, sections were incubated with secondary antibody for 1 h and with Hoechst for 10 min. Slides were mounted with Mowiol (VWR) reagent.

### Immunohistochemistry—muscle sections
Adult muscles were dissected and embedded with OCT compound. 10 µm sections were obtained and fixed in 4% paraformaldehyde for 10 min. For PAX7 staining, antigen retrieval protocol (10 mM sodium citrate buffer, pH 6) was performed. Muscle sections were blocked with 10% goat serum/0.4% Triton/PBS for 1 h and incubated with primary antibody (anti-PAX7 in 1% BSA/0.04% Triton/PBS) overnight. The next day, sections were incubated with secondary antibody Alexa Fluor (1:300) and anti-LAMININ for 1 h and with Hoechst for 10 min. For DYSTROPHIN and F4/80 staining, muscle sections were permeabilized and blocked with 1% BSA/1% goat serum/0.025% Tween 20/0.2% (Triton X-100/PBS) for 1 h and incubated with primary antibody (anti-DYSTROPHIN) overnight. The following day, sections were incubated with secondary anti-rabbit Alexa Fluor antibody with or without Alexa Fluor-conjugated anti-F4/80 for 1 h and with Hoechst for 10 min. Slides were mounted with Mowiol (VWR) reagent.

### Animal experiments
Mice were housed in a specific pathogen-free (SPF) facility and experiments were approved by the Animal Care Committee of the Institut de Recherches Cliniques de Montréal in compliance with the Canadian Council of Animal Care guidelines. *Elmo1*[-/-] mice were previously described[48]. Elmo2 KO first allele (reporter-tagged insertion with conditional potential) ES cells (clone H08) (used to generate *Elmo2*[LacZ] and *Elmo2*[flox] mice) were obtained from the International Mouse Phenotype Consortium (see Fig. 1a). *Myf5*[CRE] mice (B6.129S4-Myf5 < tm3(cre)Sor > /J), *Pax3*[CRE] mice (B6;129-Pax3[tm1(cre)Joe]/J) and *Meox2*[CRE] mice (D2.129S4(B6)-Meox2[tm1(cre)Sor/Sjl]) were from the Jackson Laboratory and were previously described[60–62]. *Elmo2*[RBD] and *Elmo2*[EID] mice were generated through homologous recombination (see targeting vector map in Supplemental Fig. 7a, b). *Dysferlin*[-/-] mice were from the Jackson Laboratory (stock #012767) (mice were kindly given by the JAIN foundation) and were previously described[52]. All genotyping primers are found in Supplemental Table 4.

### Primary myoblasts isolation and C2C12 myoblasts
C2C12 murine myoblasts were purchased from ATCC and grown and differentiated into myotubes as previously reported[15]. Experiment was performed as previously described[15]. Briefly, primary myoblasts were isolated from leg muscles of mice. Muscles were minced mechanically and digested with trypsin and collagenase D in F12 media for 1 h–3 h at 37 °C. Cells were resuspended with PBS and incubate 1 min at RT in Red Blood Cell Lysis Solution (Sigma, cat#R7757). Isolation of myoblasts was performed using magnetic beads from Miltenyl Biotec (MACS Satellite Cell Isolation Kit (cat#130-104-268) and anti-INTEGRIN-7 MicroBeads (cat#130-104-261)), where cells were incubated on ice for 15 min prior to selection on columns (LS columns; Miltenyi Biotec,

cat#130-042-401). Primary myoblasts were cultured on gelatin-coated dishes (Sigma, cat#G9382) in the following media: 39% DMEM with glutamax, 39% F12 with glutamax, 20% fetal bovine serum (Wisent) and 2% UltroserG (Pall Life Sciences, cat#15950-017). Myoblast differentiation was induced, by changing media for 2% horse serum in DMEM/F12 (with glutamax) for 72 h. Quantifications of the fusion index were performed manually by counting the number of nuclei per fiber using the Volocity software (PerkinElmer Life and Analytical Sciences).

### RNA extraction and library preparation
Primary myoblasts were isolated as described above. Seventy-two hours after differentiation, RNA was extracted using RNeasy columns following the kit protocol (QIAGEN, cat # 74104) RNA integrity was assessed on a Bioanalyzer (Agilent) and high-quality RNA samples (RNA integrity number ≥ 8) were used for library preparation. Poly-A RNAs were enriched from 1 ug total RNA using the NEBNext® Poly(A) mRNA Magnetic Isolation Module (New England Biolabs, cat #E7490) and libraries were prepared using the KAPA Stranded mRNA-Seq Kit (Roche, cat # 07962142001) with 8 cycle enrichment to minimize PCR artifacts and an average fragment size of 324 base pairs. Libraries were sequenced on a HiSeq4000 in Paired-End 50 bp fragments with an average of 52 million fragments per library.

### RNA-Seq analyses
Fastq file quality was assessed using FASTQC and aligned to the mouse genome (GRCm38) directly without trimming using STAR[63] 1-pass. Uniquely mapped reads for genes were quantified using featureCounts[64] (reverse stranded) using the ENSEMBL GTF annotations (Release 67). Differential gene expression analysis was performed with DESeq[65] (padj < 0.05). Statistical overrepresentation tests were performed with Panther, using the Panther[66] GO-Slim annotations by comparing differentially expressed genes to all expressed genes (at least one replicate with 5+ normalized counts).

### Software and codes for RNA-Seq analyses
FASTQC (0.11.5), STAR (2.5.1b), featureCounts (1.4.6), DESeq2(1.34.0), dplyr(1.0.9), ensembldb (2.18.2), ggplot2(3.3.6), ggrepel (0.9.1), GO.db (3.13.0), org.Mm.eg.db (3.13.0), pheatmap (1.0.12), PantherDB (17.0), RColorBrewer (1.1-3), reshape (0.8.9), stringr (1.4.0), svglite (2.1.0).

### Mixed myoblast population assays
Staining was performed according to the manufacturer's protocol, where primary myoblasts were stained with lipophilic cell tracking dyes PHK26 (red) or PHK67 (green) (SigmaAldrich). Briefly, following cell pellet suspension in 250ul of Diluent C, the 2x dye solution (1ul PKH in 250ul Diluent C) was added. Cells were incubated for 5 min with periodic mixing. The staining was stopped by adding an equal volume (500ul) of FBS. Finally, cell pellets were washed with F12 with glutamax media 3 times. After the last wash, the cells were resuspended in their culture media (39% DMEM with glutamax, 39% F12 with glutamax, 20% fetal bovin serum (Wisent) and 2% UltroserG (Pall Life Sciences)). Following differentiation of the two populations of myoblasts, the images were captured using the DMIRB microscope (Leica) with a 20X objective, to quantify the mixed fibers formed (see Fig. 4c). Quantification of pictures was done using the Volocity software (PerkinElmer Life and Analytical Sciences) by counting manually the number of nuclei per mixed (light pink) myofibers.

### Time-lapse
Time-Lapse movies were obtained using the DMIRE2 microscope (Leica) set up with an automated stage and a controlled environment (37 °C, 5% $CO_2$, humidity; PECON). Pictures were acquired every 10 min during 6 h at 10X. Quantification of the migration speed was done using the Volocity software (PerkinElmer Life and Analytical Sciences).

## Whole mount in situ hybridization

Whole mount in situ hybridization was performed using standard procedures[67]. Briefly, embryos were rehydrated through a methanol series (100−30%), washed in PBST (0.1% Tween) and bleached for 1 h on ice in the dark using 6% hydrogen peroxide. Embryos were treated with Proteinase K at RT for 15 min (E11.5 embryos). Following the Proteinase K treatment, embryos were fixed again with 4% paraformaldehyde (PFA). Next, embryos were hybridized with the Digoxigenin (DIG)-labelled riboprobes in hybridization buffer (5× SSC pH 4.5; 50% deionized formamide; 1% SDS; 0.1% Tween; 5 mg/mL torula RNA, 0.5 mg/mL heparin) overnight at 68 °C. Embryos were washed with 1× TBS; 0.1% Tween, treated with 10% goat serum; 1% BSA and incubated with alkaline phosphatase-conjugated anti-DIG antibodies (1/3000; Roche) overnight at 4 °C. The coloration was achieved using nitrotetrazolium blue chloride (NBT)/5-bromo-4-chloro-3-indolyl-phosphate, 4-toluidine salt (BCIP) substrate (Roche). After staining, the samples were washed in PBS and post-fixed with 4% PFA. Embryos were photographed using a DFC450.C camera mounted on a Leica, Wetzlar, Germany M165FC stereomicroscope (Wetzlar, Germany). Three embryos per genotype were assayed for reproducibility ($n = 3$). Digoxigenin (DIG)-labelled antisense riboprobes were generated from cDNA using the following primers with the reverse primers containing the T7 promoter sequence (Supplementary Table 4).

## RNA extraction, DNase treatment and reverse transcription

RNA was extracted from isolated primary myoblasts or TA muscle directly. Briefly, dissected TA muscles were flash-frozen in liquid nitrogen and crushed in powder to allow RNA extraction with TRIzol (Invitrogen) according to the manufacturer's protocol, as Qiagen's RNeasy® Mini Kit was used for RNA extraction from isolated primary myoblasts. Samples were stored at −80 °C. Total RNAs underwent DNase treatment (Invitrogen) and reverse transcription was performed with M-MuLV Reverse Transcriptase (Invitrogen) and random primers, as recommended by the manufacturer.

## Real-time PCR

All reactions were performed in a Mc3005P real-time PCR system (Stratagene) using PowerUP™ SYBR™ Green Master Mix (Applied Biosystems) in triplicate as previously described[10]. Briefly, the run started with 2 min of UDG activation at 50 °C and 2 min of Taq activation at 95 °C followed by 40 cycles of denaturation (95 °C for 15 s), primer annealing and extension (60 °C for 1 min) ending with a melting curve analysis to assess the reaction specificity. The genes investigated were mElmo1, mElmo2, mElmo3 and mB2M was used as reference gene. Primers as listed in Supplementary Table 4.

## Southern blot

The genomic probes were amplified with $^{32}P$ by PCR for use in Southern blot analysis and denatured with 1 N NaOH. Purified genomic DNA was digested overnight with the indicated restriction enzymes (see Fig. 1a and Suppl. Fig. 7a, b) and run on 0.8% agarose gel. DNA in the gel was denatured (Denaturation buffer: 1 M NaCl, 0.5 N NaOH) for 30 min. and neutralised (Neutralisation buffer: 3 M NaCl, 0.5 M Tris pH 7.5) for 30 min. Transfer was performed overnight in 10X SSC, followed by the hybridization step, where the membrane is incubated in Ambion for 1 h at 42 °C. The probe was added and left at 42 °C overnight. All washes were performed at 42 °C; gel was first rinsed with 2X SSC 0.1% SDS for 20 min, then rinsed with 0.2X SSC 0.1% SDS for 15 min and finally rinsed with 0.2X SSC.

## Protein expression and purification

The human RHOG cDNA (coding for residues 1-179) was cloned into pDEST17 using Gateway Technology, with a thrombin cleavage site inserted between the 6xHIS-tag and GTPase. The murine Elmo2 cDNA was cloned into the pGEX4T1 plasmid. The L43A mutant of Elmo2 was generated by site-directed mutagenesis. All sequences were verified by Sanger sequencing.

Non-isotopically labeled HIS-RHOG or GST-Elmo2 proteins were expressed in *Escherichia coli* BL21 codon+ cells in LB media by induction with 0.25 mM isopropyl-b-D-thiogalactopyranoside at 16 °C overnight. Isotopically labeled HIS-RHOG ($^{15}N$) was expressed in minimal M9 media supplemented with 1 g/L $^{15}NH_4Cl$. Cells were lysed via sonication in buffer comprising 20 mM Tris (pH 7.5), 150 mM NaCl, 5 mM $MgCl_2$, 10% (vol/vol) glycerol, 0.4% Nonidet P-40 and either 1 mM DTT or 5 mM β-mercaptoethanol. Lysates were cleared by centrifugation and incubated with Ni-NTA or glutathione resin for 1–2 h at 4 °C. After washing in high salt buffer (20 mM Tris (pH 7.5), 500 mM NaCl, 5 mM $MgCl_2$ and 1 mM DTT or 5 mM β-mercaptoethanol), HIS-RHOG was eluted with 250 mM imidazole followed by thrombin cleavage and the addition of 100 µM GDP. Cleaved RHOG was then further purified through a S75 size exclusion column. Cells expressing the GST-fusion Elmo2 proteins were lysed as above and bound to gluthatione sepharose for 1 h. GST-tagged Elmo2 proteins (WT or L43A mutant) were then eluted from the glutathione resin using 30 mM glutathione. Thrombin was added to the eluates overnight at 4 °C and proteins were subsequently purified through a S75 size exclusion column. For the GMPPNP nucleotide exchange of RHOG, a 10-fold molar excess of GMPPNP along with 10 mM EDTA and 1 µL of CIP (10,000 U/mL) is added for 10 min at 37 °C. 20 mM $MgCl_2$ is then added to the samples and subsequently passed through an S75 gel filtration column in 20 mM Tris, pH 7.5, 100 mM NaCl, 5 mM $MgCl_2$ and 1 mM DTT.

## NMR spectroscopy

NMR data were recorded on a 600 MHz Bruker UltraShield spectrometer at 25 °C equipped with 1.7 mm or 5 mm cryoprobes. Samples were prepared in 20 mM Tris, pH 7.5, 100 mM NaCl, 5 mM $MgCl_2$ and 1 mM DTT and 10% $D_2O$ (vol/vol). Two-dimensional $^{1}H/^{15}N$ heteronuclear single quantum coherence (HSQC) spectra were processed with NMRPIPE and analyzed using NMRView.

## Isothermal titration calorimetry

RHOG interactions with Elmo2 full-length (WT and L43A) were measured using a MicroCal ITC200 (Malvern). Experiments were conducted in 20 mM Tris, pH 7.5, 100 mM NaCl, 5 mM $MgCl_2$ and 1 mM DTT at 25 °C. For the Elmo2 full-length / RHOG interaction, 500 µM RHOG was titrated into 40 µM Elmo2 full-length (WT or L43A). Heats of dilution were determined from control experiments in which RHOG or Elmo2 were titrated into buffer alone. Datas were fit to a one-site binding model using Origin 7 (Microcal).

## Statistical analyses

Data are expressed as mean ± standard deviation from at least three independent experiments. Statistical analyses were performed with the Student's t-test (for comparison of two independent groups), using the Prism Graph software. P-value < 0.05 was considered as significant (*$P < 0.05$, **$P < 0.01$, ***$P < 0,001$, ****$P < 0.0001$).

## Structural models

The following PDB coordinates were used to build the structural models: Ternary DOCK2-ELMO1-RAC1 (6tgc); Binary DOCK2-ELMO1 (6tgb); ELMO2-BAI1 (6idx). Figures were produced using ChimeraX[68].

## Reporting summary

Further information on research design is available in the Nature Portfolio Reporting Summary linked to this article.

## Data availability

The raw RNA-seq data presented in Fig. 2h and Supplementary Fig. 9 have been uploaded to the GEO Datasets repository [https://www.ncbi.nlm.nih.gov/gds] and are available under the following accession

number: GSE209546. The remaining data are available within the article, Supplementary Information, or available from the authors upon reasonable request. Fastq file quality was aligned to the publicly available mouse genome (GRCm38). Source data are provided with this paper.

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

## Acknowledgements

We thank the JAIN foundation for the generous gift of the *Dysferlin*[KO] mouse line and helpful guidelines to analyse the dystrophic phenotype. We thank Dr. Nabil G. Seidah (IRCM) for the gift of *Meox2*[CRE] mice. We acknowledge N. Fradet for technical assistance to generate the *Elmo2* mutant mice. We recognize A. Pelletier, C. Julien and V. Luangrath for expert technical support. We thank S. Riverin for her expertise with management of the mouse colonies. We recognize the technical support and expertise of the IRCM Microinjection and transgenesis (Dr. Q. Zhu), Microscopy (Dr. D. Filion), Histology (S. Terouz) and Molecular biology, Functional genomics (Dre. Odile Neyret) and Bioinformatics (Dre. Virginie Calderon) platforms. This work was supported by a grant to J.-F.C. from the Canadian Institutes of Health Research (CIHR; PJT-153065), to M.K. (CIHR; PJT-162143), to M.J.S. (CIHR; PJT-148516), and to M.S. (CIHR; PJT-159481). V.T. (doctoral studentship), S.N. (doctoral studentship), S.E. (Doctoral studentship) and R.K. (postdoctoral fellowship) were supported by the Fonds de Recherche du Québec-Santé (FRQS). I.D. was supported by an IRCM-Jean Coutu fellowship from the IRCM Foundation. M.S. is a FRQS Research Scholar Junior 1. M.J.S. holds a Canada Research Chair in Cancer Signaling and Structural Biology. J.-F.C. holds the Canada Research Chair in Cellular Signaling and Cancer Metastasis and the Alain Fontaine chair in cancer research (IRCM Foundation).

## Author contributions

V.T. and J.-F.C. designed the research; V.T., S.N., A.R., I.D., R.K., S.E. and M.P.T. performed the research; K.S.R. provided Elmo1 mice; D.B. generated structural models; V.T., S.N., A.R., I.D., R.K., S.E., M.S., M.J.S., M.K. and J.-F.C. analyzed the data. V.T., A.R. and J.-F.C. wrote the paper with input from all authors.

## Competing interests

The authors declare no competing interests.
