## [Peer Review File · Nature Communications]

Biasing the conformation of ELMO2 reveals that myoblast fusion can be exploited for regenerative therapyReviewer #1 (Remarks to the Author):

Review: Biasing the conformation of Elmo2 reveals that myoblasts fusion can be exploited for regenerative therapy (by Tran et al).

In this study, the authors examined the role(s) of Elmo proteins, Dock1-interacting scaffolds, in myoblast fusion, a process critical for the formation of multinucleated fibers during skeletal muscle development and regeneration. The motivation for doing so is based on their previous findings demonstrating that Rac1 and its activator Dock1 are essential for proper myoblast fusion. The Elmo proteins act in concert with Dock1 to activate Rac1 GTPase, and this complex is tightly regulated by auto-inhibitory mechanisms. A mutation in the N-terminally located Elmo inhibitory domain (EID) releases the inhibitory contacts, leading to enhanced signaling of the complex, whereas a mutation in the Ras-binding domain (RBD) of Elmo abolishes its function. The authors began by examining the expression profiles of all three Elmo genes (Elmo1, 2, and 3) and found all three genes to be expressed in somites, the sites of myogenesis, as well as in the adult murine tibialis anterior (TA) muscle and primary myoblasts. They then went on to generate a series of Elmo1 and Elmo2 single and double knockout (KO) animals, with Elmo2 being conditionally ablated in the muscle lineage (as global deletion of Elmo1 and Elmo2 leads to early embryonic lethality). While Elmo1 deletion or muscle-specific Elmo2 deletion alone had no effect, analyses of embryos of double homozygous animals (Elmo1KO;Myf5fCREElmo2flox/flox and Elmo1KO;Pax3CREElmo2flox/flox) revealed the presence of mononucleated myofibers and a reduction in muscle content, indicating that Elmo1 and 2 have a redundant function that is essential for proper embryonic myogenesis. Next, the authors generated Elmo2 knock-in mice carrying a mutation in either the RBD or EID domain and found that Elmo1-/-;Elmo2RBD/RBD mice exhibited smaller muscle fibers and decreased myoblast fusion compared to control mice, while Elmo2EID/EID mice displayed larger myofibers and a higher number of nuclei per fiber. Interestingly, similar results were obtained in the context of muscle regeneration, with the Elmo2RBD mutation causing less efficient muscle regeneration and the Elmo2EID mutation increased myoblast fusion events in the regenerating muscles. Finally, the authors presented data demonstrating that expression of Elmo2EID in Dysferlin KO mice is able to ameliorate several of the dystrophic features of these mice.

Overall, this is an interesting study unveiling a critical role for Elmo1 and Elmo2 in myoblast fusion during development and in muscle regeneration. The manuscript is well written and the experiments are generally carefully executed. A number of points the authors should address are listed below.

Specific points

- 1) The authors provide data/images supporting that double homozygous Elmo1KO;Myf5fCREElmo2flox/flox and Elmo1KO;Pax3CREElmo2flox/flox embryos display mononucleated myofibers and a reduction in muscle content compared to wild-type embryos. To obtain a better appreciation of the extent of myoblast fusion defects, quantifications for Fig. 1c and d should be included. Similarly, quantifications should be included for Fig 1g and Fig. S6a – which evaluate differentiation defects.**
- 2) The authors show that 14 and 21 days following cardiotoxin (CTX)-induced injury Elmo1KO;ElmoRBD/RBD mice display a smaller muscle cross-sectional area (CSA) and a decrease in the size of myofibers compared to control mice. On the other hand, Elmo2EID/EID mice exhibited a larger CSA area and a higher number of myofibers with three nuclei - 14 and 21 days following CTX-induced injury. These data indicate that muscle regeneration is less efficient when signaling by the RBD of Elmo2 is disrupted while Elmo2 in its open configuration (Elmo2EID) promotes muscle regeneration. Does expression of Elmo2EID also accelerates the muscle regeneration process? For instance, can regeneration be seen one week following CTX-induced injury whilst this not being**

the case for control conditions? Also, the authors should include images of tibialis anterior (TA) muscles immediately following CTX-induced injury (i.e. before regeneration occurs).

3) The authors demonstrate that Elmo2 in its open conformation (Elmo2EID) increases myoblast fusion. They further show that Elmo2EID does not significantly impact satellite cell proliferation, myoblast differentiation and migration nor the expression of fusogens. Do the authors have any insight as to how Elmo2EID promotes myoblast fusion? As cell migration (which is a Rac1-dependent process) is not affected, did the authors check the levels of active Rac1 in Elmo2EID-expressing myoblasts? Along the same lines, the authors should assess whether the actin cytoskeleton is altered in these cells.

4) The finding that expression of Elmo2EID can ameliorate several of the dystrophic features of Dysferlin-null mice is interesting. The authors indicate in several places throughout the manuscript that such findings could be exploited for regenerative and therapeutic purposes. The authors should elaborate on this in the discussion. Is there a molecule/peptide that can trigger the open configuration of Elmo2? Or can such a compound be generated? In this regard, it will be important to know whether expression of Elmo2EID triggers a major cellular effect beyond that seen in muscles. Do they have any data to suggest this?

Reviewer #2 (Remarks to the Author):

This study explores the dependence of fusion of mammalian myoblasts *in vivo* and *ex vivo* on the expression and activity of Elmo1 and Elmo2, regulators of actin-mediated processes. While mice deficient in Elmo1 has no obvious muscle phenotype, mice deficient in both Elmo1 and, specifically in skeletal muscles, in Elmo2 demonstrate myoblast fusion defects. Elmo1 and 2 are found to have redundant functions. To manipulate the conformational state of Elmo2 in mice, the authors developed and characterized Elmo2 mutants with locked open (Elmo2EID, promotes myoblast fusion) and closed (Elmo2RBD, inhibits myoblast fusion) conformations. While these findings are solid, the mechanisms underlying the Elmo-dependence of muscle development remain to be further clarified. The paper also reports a potentially important finding that the Elmo2 mutant with locked open conformation rescues fusion defect in the Dysferlin-deficient mice. The results are novel and the work is well-written and illustrated but in my opinion can be further strengthened by several additional experiments solidifying some of the key conclusions of the study.

Specific comments.

1) Actin rearrangements obviously play important role in diverse cell biological processes. Assuming that primary myoblasts have been isolated from adult animals (as in cited paper Ref 15), the long-term major changes in the expression and properties of actin regulators Elmo proteins may cause compensating changes in the expression and function of other proteins. The hypothesis that Elmo2EID directly promotes fusion of dysferlin-deficient myoblasts can be supported by finding that overexpressing Elmo2EID in primary myoblasts from dysferlin-deficient mice promotes their fusion.

2) Are there any changes in the expression of Elmo3 in Elmo1 and Elmo2 deficient mice?

3) Finding that Myomaker and Myomixer mRNA levels were unchanged following differentiation of Elmo2EID myoblasts (Fig. 4f) has to be supported by Westerns.

4) Is myogenic differentiation of wt myoblasts *in vitro* accompanied by any changes in expression of Elmo proteins?

5) The differences in the rates of muscle regeneration after microinjury can reflect not only differences in the rates of myogenic differentiation/fusion but also differences in the injury and post-injury inflammation (as noted by the authors in the discussion). Are the numbers of macrophages (that can be detected as cells positive for the pan-macrophage marker F4/80) per given area at early time after the injury similar for wt and mutant mice (Elmo1-/Elmo2 RBD/RBD and Elmo2EID/EID)?

6) Finding that expression of Elmo2EID/EID in dysferlin-deficient mice decreases the percentage of centrally nucleated myofibers (Fig. 5f) is very intriguing. Does it suggest that Elmo 2 is involved in post-fusion stages of myogenesis? How do the authors interpret this finding?

7) To substantiate the hypothesis that the effects of RBD and EID mutations influence fusion by locking open/closed conformations of ELMO2, it is important to verify by Westerns that these mutations do not change the levels of expression/turn-over of the protein.

8) Can the contributions of Elmo2 to myoblast fusion reflect the known importance of this protein in the development of strong cell-cell contacts (Toret et al., PMID: PMC4259811)?

9) Can the finding that Elmo2EID/EID rescues myoblast fusion of Dysferlin-deficient myoblasts be explained by a suppressed Elmo signaling in Dysferlin-null mice? It can be interesting to compare the levels of expression of Elmo2 in primary myoblasts from Dysferlin-null mice, wild type mice and Dysferlin-/-Elmo2EID/EID mice.

Minor.

I have noted some typos: "In contrats to Myomaker"; "such that smaller and damages muscle fibers"; "in vertabrates"; "Elmo2EID-dependant increase"; "lead to the lose of some".

REBUTTAL LETTER

We thank the editor and the expert reviewers for their overall positive comments of our manuscript. As requested, we have conducted significant amount of new work to improve our manuscript. We believe that we have addressed all the constructive comments of the two reviewers below.

Reviewer #1

“In this study, the authors examined the role(s) of Elmo proteins, Dock1-interacting scaffolds, in myoblast fusion, a process critical for the formation of multinucleated fibers during skeletal muscle development and regeneration. The motivation for doing so is based on their previous findings demonstrating that Rac1 and its activator Dock1 are essential for proper myoblast fusion. The Elmo proteins act in concert with Dock1 to activate Rac1 GTPase, and this complex is tightly regulated by auto-inhibitory mechanisms. A mutation in the N-terminally located Elmo inhibitory domain (EID) releases the inhibitory contacts, leading to enhanced signaling of the complex, whereas a mutation in the Ras-binding domain (RBD) of Elmo abolishes its function. The authors began by examining the expression profiles of all three Elmo genes (Elmo1, 2, and 3) and found all three genes to be expressed in somites, the sites of myogenesis, as well as in the adult murine tibialis anterior (TA) muscle and primary myoblasts. They then went on to generate a series of Elmo1 and Elmo2 single and double knockout (KO) animals, with Elmo2 being conditionally ablated in the muscle lineage (as global deletion of Elmo1 and Elmo2 leads to early embryonic lethality). While Elmo1 deletion or muscle-specific Elmo2 deletion alone had no effect, analyses of embryos of double homozygous animals (Elmo1KO;Myf5CRE Elmo2flox/flox and Elmo1KO;Pax3CREElmo2flox/flox) revealed the presence of mononucleated myofibers and a reduction in muscle content, indicating that Elmo1 and 2 have a redundant function that is essential for proper embryonic myogenesis. Next, the authors generated Elmo2 knock-in mice carrying a mutation in either the RBD or EID domain and found that Elmo1-/-;Elmo2RBD/RBD mice exhibited smaller muscle fibers and decreased myoblast fusion compared to control mice, while Elmo2EID/EID mice displayed larger myofibers and a higher number of nuclei per fiber. Interestingly, similar results were obtained in the context of muscle regeneration, with the Elmo2RBD mutation causing less efficient muscle regeneration and the Elmo2EID mutation increased myoblast fusion events in the regenerating muscles. Finally, the authors presented data demonstrating that expression of Elmo2EID in Dysferlin KO mice is able to ameliorate several of the dystrophic features of these mice.

Overall, this is an interesting study unveiling a critical role for Elmo1 and Elmo2 in myoblast fusion during development and in muscle regeneration. The manuscript is well written and the experiments are generally carefully executed. A number of points the authors should address are listed below.”

We thank this reviewer for the positive assessment of our manuscript. Below, we address all the constructive concerns raised.

Specific points:

“1) The authors provide data/images supporting that double homozygous Elmo1KO;Myf5fCREElmo2flox/flox and Elmo1KO;Pax3CREElmo2flox/flox embryos display mononucleated myofibers and a reduction in muscle content compared to wild-type embryos. To obtain a better appreciation of the extent of myoblast fusion defects, quantifications for Fig. 1c and d should be included. Similarly, quantifications should be included for Fig 1g and Fig. S6a – which evaluate differentiation defects.”

This is an excellent point. We now provide quantification of Fig. 1g (*see new Fig. 1h*) and Supplemental Fig. 6a (*see new Supplemental Fig. 6b*), as requested. The approach used for this quantification is described in the method section.

However, Fig 1c was virtually impossible to quantify as it is difficult to track individual fibres. We can select the best longitudinal sections to stain muscle fibres when myoblast fusion is not affected (first 3 panels of Fig 1c), but not in ELMO1 and ELMO2 double mutants (last 2 panels). Because of the severe defects in fusion in these conditions, it is more challenging to know if the tissue sections are identical to the control ones. However, all sections obtained in double mutant embryos show the strong phenotype of myoblast fusion defects with most cells, as far as we can see in such an assay, are mono-nucleated. Almost identical robust phenotypic observations of impaired primary myoblast fusion were reported for RAC1/CDC42¹, N-WASP², MYMK³, MYMX^{4,5} and DOCK1/DOCK5⁶ mutants and these experiments were not quantified. While we agree that quantified data would be ideal, we believe that our experiment clearly demonstrates the major defect in myoblast fusion when ELMO1/2 proteins are eliminated during embryogenesis.

2) The authors show that 14 and 21 days following cardiotoxin (CTX)-induced injury Elmo1KO;ElmoRBD/RBD mice display a smaller muscle cross-sectional area (CSA) and a decrease in the size of myofibers compared to control mice. On the other hand, Elmo2EID/EID mice exhibited a larger CSA area and a higher number of myofibers with three nuclei - 14 and 21 days following CTX-induced injury. These data indicate that muscle regeneration is less efficient when signaling by the RBD of Elmo2 is disrupted while Elmo2 in its open configuration (Elmo2EID) promotes muscle regeneration. Does expression of Elmo2EID also accelerates the muscle regeneration process? For instance, can regeneration be seen one week following CTX-induced injury whilst this not being the case for control conditions? Also, the authors should include images of tibialis anterior (TA) muscles immediately following CTX-induced injury (i.e. before regeneration occurs).

This is an excellent suggestion. First, we have now included the control conditions 1 day after CTX-induced injury (*Supplemental Fig. 10a*). We also analyzed *Elmo2^{EID}* mice at 7 days post CTX-induced injury and quantified CSA and the numbers of centrally localized nuclei in comparison to WT mice (*Supplemental Fig. 10a-d*). These experiments demonstrate that at 7 days post-injury, muscles look identical between WT and *Elmo2^{EID}* mice suggesting that the regeneration process does not start earlier or is not accelerated. We also invite the reviewer to read our answer to the point 5 of reviewer 2 who is asking to investigate if macrophage recruitment is altered in our *Elmo2^{EID}* model. We provide evidence that this is not the case (see *Supplementary Fig. 10e*).

“3) The authors demonstrate that Elmo2 in its open conformation (Elmo2EID) increases myoblast fusion. They further show that Elmo2EID does not significantly impact satellite cell proliferation, myoblast differentiation and migration nor the expression of fusogens. Do the authors have any insight as to how Elmo2EID promotes myoblast fusion? As cell migration (which is a Rac1-dependent process) is not affected, did the authors check the levels of active Rac1 in Elmo2EID-expressing myoblasts? Along the same lines, the authors should assess whether the actin cytoskeleton is altered in these cells.”

First, to provide context, we previously reported the existence of a conformational switch in ELMO proteins (see ⁷), we extensively characterized the effect of over-expressing ELMO1^{RBD} or ELMO1^{EID} mutant proteins in fibroblasts. We found that ELMO1^{EID} could induce an elongated cell shape in about 50% of expressing cells. However, measuring RAC1 activity (PAK pull-down assay) did not show global changes in RAC1 activation suggesting that a small pool of RAC1 is being activated. We used a RAC1 Raichu FRET probe to demonstrate that ELMO1^{EID} can promote a localized small increase in Rac1 activity where ELMO1^{EID} is localized ⁸. Finally, in a collaboration with Dr. David Barford's laboratory, we demonstrated that purified EILMO1/DOCK2 complex was minimally active in a RAC1 GEF assay while a

complex composed of ELMO1 lacking the auto-inhibitory region and DOCK2 showed GEF activity⁹. These data support a mechanism whereby release of ELMO/DOCK auto-inhibition will allow localized RAC1 activation.

We agree with the experiments requested by the reviewer. However, based on our experiences of over-expression in fibroblasts, we did not anticipate that the endogenous ELMO2^{EID} expression in our mouse model would be sufficient to induce a phenotype such as constitutive global high RAC1 activation. First, we isolated primary myoblasts from WT and *Elmo2*^{EID} mice and captured several images in the early phase of differentiation. We report in new **Supplemental Fig. 12a-f** that myoblasts of both genotypes show similar actin cytoskeleton organization. We also conducted several RAC1 activity assays on primary myoblasts (PAK pull-down kit from Cytoskeleton Inc) and we failed to detect significant and/or reproducible differences in RAC1-GTP loading between WT or *Elmo2*^{EID} myoblasts in proliferation or in differentiation. We decided not to include this data to avoid over-loading the manuscript with negative data. *These observations are consistent with a model where ELMO2^{EID} does not globally activate RAC1 in cells, but instead, the ELMO2^{EID} mutant is likely for primed RAC1 activation where it localizes.*

Reviewer 2 also raised a similar point by stating that constitutive expression of ELMO2^{EID} in mice may lead to compensation mechanisms or altered transcriptomes. We address this in our response to point #1 of reviewer 2. Briefly, we conducted transcriptomics analyses of WT and *Elmo2*^{EID} myoblasts undergoing fusion and found that the core myoblast fusion machinery genes are not strikingly differentially expressed. Please see our full answer to his comment below.

“4) The finding that expression of Elmo2EID can ameliorate several of the dystrophic features of Dysferlin-null mice is interesting. The authors indicate in several places throughout the manuscript that such findings could be exploited for regenerative and therapeutic purposes. The authors should elaborate on this in the discussion. Is there a molecule/peptide that can trigger the open configuration of Elmo2? Or can such a compound be generated? In this regard, it will be important to know whether expression of Elmo2EID triggers a major cellular effect beyond that seen in muscles. Do they have any data to suggest this?”

This is an excellent suggestion. We propose a model to derive biologically active peptides from the regions of either BAI1, BAI2 or BAI3 GPCRs that specifically bind ELMO proteins. Such a fragment of BAI1 has recently been crystalized bound to the ELMO2 N-terminus¹⁰. Here, we overlay the BAI1 fragment onto the structure of open and closed conformation of the DOCK2-ELMO1 complex⁹. We demonstrate with these structural models that such fragments of BAI-receptors, when bound to ELMO proteins, would be incompatible with the auto-inhibited state of the ELMO-DOCK complex. These data are presented in **Supplementary Fig. 13**. As such, we discuss the possibility of using a BAI-derived peptide to activate ELMO-DOCK in future studies.

Reviewer #2:

“This study explores the dependence of fusion of mammalian myoblasts in vivo and ex vivo on the expression and activity of Elmo1 and Elmo2, regulators of actin-mediated processes. While mice deficient in Elmo1 has no obvious muscle phenotype, mice deficient in both Elmo1 and, specifically in skeletal muscles, in Elmo2 demonstrate myoblast fusion defects. Elmo1 and 2 are found to have redundant functions. To manipulate the conformational state of Elmo2 in mice, the authors developed and characterized Elmo2 mutants with locked open (Elmo2EID, promotes myoblast fusion) and closed (Elmo2RBD, inhibits myoblast fusion) conformations. While these findings are solid, the mechanisms underlying the Elmo-dependence of muscle development remain to be further clarified. The paper also reports a potentially important finding that the Elmo2 mutant with locked open conformation rescues fusion defect in the Dysferlin-deficient mice. The results are novel and the work is well-written and

illustrated but in my opinion can be further strengthened by several additional experiments solidifying some of the key conclusions of the study.”

We thank the reviewer for the positive comments on our study and the potential novelty and importance of our findings. Point by point answers to all Reviewer 2’s comment are provided below.

Specific comments

“1) Actin rearrangements obviously play important role in diverse cell biological processes. Assuming that primary myoblasts have been isolated from adult animals (as in cited paper Ref 15), the long-term major changes in the expression and properties of actin regulators Elmo proteins may cause compensating changes in the expression and function of other proteins. The hypothesis that Elmo2^{EID} directly promotes fusion of dysferlin-deficient myoblasts can be supported by finding that overexpressing Elmo2^{EID} in primary myoblasts from dysferlin-deficient mice promotes their fusion.”

Indeed, WT and *Elmo2^{EID}* myoblasts were isolated from adult mice, and we agree with the reviewer that there might be compensatory mechanisms at play to explain the difference in fusion between these myoblast genotypes. Despite this caveat, we believe that crossing *Dysf-null* mice with a mouse model where endogenous mutation in ELMO2 (EID) is present represent a cleaner genetic experiment than conducting over-expression assays of ELMO2^{EID}. Our previous work showed that over-expression of ELMO1^{EID} in fibroblasts required titration of an optimal expression level to detect biological effects (see ⁷). Also, the recent Cryo-EM structure revealed that ELMO proteins must be in complex with a DOCK protein for their functions (see our collaborative effort ⁹) – such that over-expression would result in a lot of “free” ELMO2^{EID}. Therefore, we were reluctant to transfect/infect *Dysf-null* myoblast which would be challenging since it would be difficult to obtain the needed expression levels of ELMO2^{EID} while out-competing endogenous ELMO1,2,3 to bind DOCK.

Instead, to directly address the question of compensation and transcriptomics changes in myoblasts expressing ELMO2^{EID}, we isolated and differentiated (72 hours) primary myoblasts from WT (n=3) and *Elmo2^{EID}* (n=2) animals and next conducted an RNA-seq and transcriptomics analyses. We found that 961 genes were differentially expressed. There was enrichment in GO terms including “extracellular matrix”, “chemotaxis”, and “cell-cell adhesion” (see **Supplemental Fig. 9**). Next, to directly address the reviewer’s comment, we asked whether genes from the “core myoblast fusion” machinery were differentially expressed. We generated a list of 30 genes from the literature to be involved in myoblast fusion (27 were expressed in differentiated myoblasts). We found three significantly differentially expressed genes: *Pak1*, *Tgfb2* and *Dysf* (see **Fig. 2h**). *These are minimal changes in the myoblast fusion machinery, and we conclude that the increased fusion observed in the Elmo2^{EID} myoblast is likely due to the biochemical properties of this mutant form of ELMO2, but we are careful not to completely rule out transcriptional changes.* We also included a brief discussion on the potential roles of these dysregulated genes on fusion in the discussion (*Tgfb2* and *Dysf*). Notably, (i) downregulation of TGFBR2 is consistent with a regulatory role of TGF-beta signalling in pacing cell-cell fusion; (ii) increase expression of *Dysf* in *Elmo2^{EID}* myofibers is intriguing and will be the focus of a future study. Notably, *Mymk* and *Mymx* are unchanged.

“2) Are there any changes in the expression of Elmo3 in Elmo1 and Elmo2 deficient mice?”

This is an excellent point. We measure the expression levels of *Elmo3* in *Myf5^{CRE}Elmo1^{-/-}Elmo2^{Flox/Flox}* in comparison to WT tissues of E14.5 embryos (limb and surrounding tissues). We found that *Elmo3* expression is not different between these genotypes ruling out a compensatory function of ELMO3 to the observed phenotypes. This new data is presented in **Supplemental Fig. 5e**. We also found from the RNA-Seq data discussed above (point 1) that *Elmo3* expression is unaffected in *Elmo2^{EID}* myoblasts in comparison to control myoblasts (not shown; data available on GEO). *In summary, based on these data, it*

is unlikely that altered expression of *Elmo3* would contribute to the phenotypes observed in this study when *Elmo1/2* are manipulated.

“3) Finding that *Myomaker* and *Myomixer* mRNA levels were unchanged following differentiation of *Elmo2^{EID}* myoblasts (Fig. 4f) has to be supported by Westerns.”

We agree that this would be the ideal proof. However, it is challenging to detect endogenous MYMK and MYMX protein with current available antibodies. We obtained a collection of antibodies (commercial and from various labs including, Doug Millay and Leonid Chernomordik). While we could detect over-expressed proteins, and to some extent also endogenous proteins in C2C12, we failed to obtain consistent results on primary myoblasts. We show the inconsistent data obtained here. The data goes along the same directions as the mRNA expression levels of *Mymk* and *Mymx*, but since we do not fully trust the Western blot signals, we prefer not to include it in the paper. Also, the new RNA-Seq experiments shown in Fig. 2h shows there is no differences in *Mymk* and *Mymx* expression between WT and *Elmo2^{EID}* differentiated myoblasts.

“4) Is myogenic differentiation of wt myoblasts in vitro accompanied by any changes in expression of *Elmo* proteins?”

To address this comment, we conducted qRT-PCR on RNA isolated from C2C12 cells in proliferation or that differentiated for 24, 48 or 72 hours. Our results demonstrate that expression of *Elmo* genes is not significantly modified during differentiation (see Supplementary Fig. 1d). By conducting classical RT-PCR analyses, our laboratory previously demonstrated that the levels of *Elmo1* and *Elmo2* were not modulated during differentiation of C2C12 cells and that *Elmo3* was barely detectable in this assay.

“5) The differences in the rates of muscle regeneration after microinjury can reflect not only differences in the rates of myogenic differentiation/fusion but also differences in the injury and post-injury inflammation (as noted by the authors in the discussion). Are the numbers of macrophages (that can be detected as cells positive for the pan-macrophage marker F4/80) per given area at early time after the injury similar for wt and mutant mice (*Elmo1*-/*Elmo2* RBD/RBD and *Elmo2^{EID/EID}*)?”

This is an excellent point that we indeed raised ourselves in the discussion of our original submitted manuscript. We focused on the *Elmo2^{EID}* mouse model for these experiments. For technical reasons of significant downscaling of our mouse colonies last fall due to the COVID pandemic and pressure on our animal facilities, we have been unable to restart efficient breeding of the *Elmo1*^{-/-}*Elmo2^{RBD}* line (luckily sperm is frozen). Nevertheless, we felt that the *Elmo2^{EID}* model was more interesting to characterize in that respect since improved regeneration is observed and this could offer a potential mechanism. We analyzed macrophage infiltration (anti-F4/80 staining) 3 days post CTX-induced injuries in n=3 mice. We found no

major difference in the quantity of macrophages present in injured tissues at that time (see new **Supplemental Fig. 10e**). Hence, we conclude that it is unlikely that increased recruitment of macrophages in regenerating muscles tissues of *Elmo2^{EID}* mice would explain the improved regeneration observed. This is also included in the discussion (see page 16).

“6) Finding that expression of Elmo2EID/EID in dysferlin-deficient mice decreases the percentage of centrally nucleated myofibers (Fig. 5f) is very intriguing. Does it suggest that Elmo 2 is involved in post-fusion stages of myogenesis? How do the authors interpret this finding?”

Our interpretation of this data is that the muscles in the *Dysf-null* mice are undergoing constant degeneration and that myoblast fusion is occurring for regeneration. The presence of centrally localized nuclei fits with recent myoblast fusion events. In the experiment presented in Fig.5 f, these mice were aged between 8-12 months such that fusion has occurred over a long time. In the *Elmo2^{EID}* model, we propose that there is less requirement for myoblast fusion and hence less detection of centrally localized nuclei. We now discuss this interpretation in the discussion section.

“7) To substantiate the hypothesis that the effects of RBD and EID mutations influence fusion by locking open/closed conformations of ELMO2, it is important to verify by Westerns that these mutations do not change the levels of expression/turn-over of the protein.”

This is an excellent comment. We isolated TA muscles from WT, *Elmo2^{RBD}* and *Elmo2^{EID}* to address this question. We extracted proteins and monitored ELMO2 expression by Western blot, which confirmed that the RBD and EID mutation do not alter the expression levels of these proteins. This new data is presented and quantified in **Supplemental Fig. 7f-g**.

“8) Can the contributions of Elmo2 to myoblast fusion reflect the known importance of this protein in the development of strong cell–cell contacts (Toret et al., PMID: PMC4259811)?”

We thank the reviewer for pointing to this paper that is indeed potentially relevant in the context of myoblast fusion. The roles of ELMO and DOCK proteins in promoting strong Cadherin-mediated cell-cell contacts have been demonstrated in *Drosophila* S2 cells, *in vivo* in fly embryos during dorsal closure, as well as in an MDCK cell model. Whether this function is conserved in myoblast is unknown. Alternatively, *Elmo2^{EID}* myoblasts may show increased expression of cell-cell adhesion genes. Indeed, we found 6 genes showing moderate increase in expression in *Elmo2^{EID}* myoblasts that belong to the GO term “cell-cell adhesion”. We added a section in the discussion that this could be an interesting area of investigation to pursue to further understand the role of the ELMO/DOCK complex in myoblast fusion.

“9) Can the finding that Elmo2EID/EID rescues myoblast fusion of Dysferlin-deficient myoblasts be explained by a suppressed Elmo signaling in Dysferlin-null mice? It can be interesting to compare the levels of expression of Elmo2 in primary myoblasts from Dysferlin-null mice, wild type mice and Dysferlin-/-Elmo2EID/EID mice.”

We monitored the levels of ELMO2 in WT, *Dysf-null* and *Dysf-null:Elmo2^{EID}* myoblasts and found that ELMO2 is expressed similarly in all genotypes, with slightly less in *Dysf-null:Elmo2^{EID}* cells. This new data is added as **Fig. 5 k**. In the revised manuscript, we conclude that ELMO2 levels are not significantly altered in myoblasts from these mouse models.

Minor

“I have noted some typos: “In contrats to Myomaker”; “such that smaller and damages muscle fibers”; “in vertebrates”; “Elmo2EID-dependant increase”; “lead to the lose of some”.”

We have secured the help of a professional editor to help correct English mistakes as much as possible.

REFERENCES

- 1 Vasyutina, E., Martarelli, B., Brakebusch, C., Wende, H. & Birchmeier, C. The small G-proteins Rac1 and Cdc42 are essential for myoblast fusion in the mouse. *Proc Natl Acad Sci U S A* **106**, 8935-8940, doi:0902501106 [pii] 10.1073/pnas.0902501106 (2009).
- 2 Gruenbaum-Cohen, Y. *et al.* The actin regulator N-WASp is required for muscle-cell fusion in mice. *Proc Natl Acad Sci U S A* **109**, 11211-11216, doi:10.1073/pnas.1116065109 (2012).
- 3 Millay, D. P. *et al.* Myomaker is a membrane activator of myoblast fusion and muscle formation. *Nature* **499**, 301-305, doi:10.1038/nature12343 (2013).
- 4 Bi, P. *et al.* Control of muscle formation by the fusogenic micropeptide myomixer. *Science* **356**, 323-327, doi:10.1126/science.aam9361 (2017).
- 5 Quinn, M. E. *et al.* Myomerger induces fusion of non-fusogenic cells and is required for skeletal muscle development. *Nature communications* **8**, 15665, doi:10.1038/ncomms15665 (2017).
- 6 Laurin, M. *et al.* The atypical Rac activator Dock180 (Dock1) regulates myoblast fusion in vivo. *Proc Natl Acad Sci U S A* **105**, 15446-15451, doi:0805546105 [pii] 10.1073/pnas.0805546105 (2008).
- 7 Patel, M. *et al.* An evolutionarily conserved autoinhibitory molecular switch in ELMO proteins regulates Rac signaling. *Curr Biol* **20**, 2021-2027, doi:S0960-9822(10)01292-3 [pii] 10.1016/j.cub.2010.10.028 (2010).
- 8 Margaron, Y., Fradet, N. & Cote, J. F. ELMO Recruits Actin Cross-linking Family 7 (ACF7) at the Cell Membrane for Microtubule Capture and Stabilization of Cellular Protrusions. *J Biol Chem* **288**, 1184-1199, doi:10.1074/jbc.M112.431825 (2013).
- 9 Chang, L. *et al.* Structure of the DOCK2-ELMO1 complex provides insights into regulation of the auto-inhibited state. *Nature communications* **11**, 3464, doi:10.1038/s41467-020-17271-9 (2020).
- 10 Weng, Z. *et al.* Structure of BAI1/ELMO2 complex reveals an action mechanism of adhesion GPCRs via ELMO family scaffolds. *Nature communications* **10**, 51, doi:10.1038/s41467-018-07938-9 (2019).

Reviewer #1 (Remarks to the Author):

In the revised manuscript, the authors have addressed most of my previously brought-up concerns, either by including additional experimental data or providing better explanations. The current manuscript provides significant novel information (beyond previous studies) that I believe will be of great interest to a large readership. And therefore, I consider the current version suitable for publication in Nature Communications.

Reviewer #2 (Remarks to the Author):

The authors have addressed my comments in a satisfactory manner. The evidence justifies the paper's conclusions, and the contributions of the DOCK1-interacting ELMO proteins to myoblast fusion will be of importance for the field.

Response to reviewer letter

“Dear Dr Côté,

Your manuscript entitled "Biasing the conformation of ELMO2 reveals that the myoblast fusion process can be exploited for regenerative therapy" has now been seen again by our referees, whose comments appear below. In light of their advice I am delighted to say that we are happy, in principle, to publish a suitably revised version in Nature Communications under the open access CC BY license (Creative Commons Attribution 4.0 International License).”

We thank the editor for the excellent handling of our manuscript throughout the review process.

Reviewer #1

“In the revised manuscript, the authors have addressed most of my previously brought-up concerns, either by including additional experimental data or providing better explanations. The current manuscript provides significant novel information (beyond previous studies) that I believe will be of great interest to a large readership. And therefore, I consider the current version suitable for publication in Nature Communications.”

We thank this reviewer for the positive assessment of our revised manuscript.

Reviewer #2:

“The authors have addressed my comments in a satisfactory manner. The evidence justifies the paper's conclusions, and the contributions of the DOCK1-interacting ELMO proteins to myoblast fusion will be of importance for the field.”

We thank the reviewer for the previous constructive feedbacks and for reviewing our manuscript again.